# Brain-derived exosomal hemoglobin transfer contributes to neuronal mitochondrial homeostasis under hypoxia

Zhengming Tian[1], Yuning Li[1], Feiyang Jin[1,2], Zirui Xu[1], Yakun Gu[1], Mengyuan Guo[1], Qianqian Shao[1], Yingxia Liu[1], Hanjiang Luo[3], Yue Wang[4], Suyu Zhang[4], Chenlu Yang[4], Xin Liu[4], Xunming Ji[1,2]*, Jia Liu[1]*

[1]Beijing Institute of Brain Disorders, Laboratory of Brain Disorders, Laboratory for Clinical Medicine, Chinese Institutes for Medical Research, Ministry of Science and Technology, Collaborative Innovation Center for Brain Disorders, Beijing Advanced Innovation Center for Big Data-based Precision Medicine, Capital Medical University, Beijing, China; [2]Department of Neurosurgery, Xuanwu Hospital, Capital Medical University, Beijing, China; [3]Laboratory of Neuroscience, Affiliated Hospital of Guilin Medical University, Guilin, China; [4]BGI-Beijing, Beijing, China

*For correspondence:
jixm@ccmu.edu.cn (XJ);
liujia_19901005@163.com (JL)

Competing interest: The authors declare that no competing interests exist.

## eLife Assessment

This **valuable** article analyzes the role of endogenous CNS hemoglobin in protecting mitochondrial homeostasis in hypoxic conditions. The work is **solid** and opens the doors to future work in this field. However, it leaves many questions open regarding CNS-specific ischemia/hypoxia that should be considered in future work. In particular, a whole-body hypoxia model may liberate exosomes from other hypoxic organs, which may contribute to the protective effect. Overall, this work has the potential to be of broad interest to the neuroscience and hypoxia communities.

**Abstract** Hypoxia is an important physiological stress causing nerve injuries and several brain diseases. However, the mechanism of brain response to hypoxia remains unclear, thus limiting the development of interventional strategies. This study conducted combined analyses of single-nucleus transcriptome sequencing and extracellular vesicle transcriptome sequencing on hypoxic mouse brains, described cell–cell communication in the brain under hypoxia from intercellular and extracellular dimensions, confirmed that hemoglobin mRNA was transferred from non-neuronal cells to neurons, and eventually expressed. Then we further explored the role of exosomal hemoglobin transfer in vitro, using human-derived cell lines, and clarified that hypoxia promoted the transfer and expression of exosomal hemoglobin between endothelial cells and neurons. We found the vital function of exosomal hemoglobin to protect against neurological injury by maintaining mitochondrial homeostasis in neurons. In conclusion, this study identified a novel mechanism of 'mutual aid' in hypoxia responses in the brain, involving exosomal hemoglobin transfer, clarified the important role of exosomal communication in the process of brain stress response, and provided a novel interventional perspective for hypoxia-related brain diseases.

## Introduction

Extracellular vesicles (EVs) are membrane-enclosed vesicles secreted by cells into the extracellular environment and are an important means of cell communication (*van Niel et al., 2022*; *Liang*

*et al., 2023*; *Cocozza et al., 2023*). EV-mediated communication can participate in the regulation of intercellular signals in physiological and pathological conditions by transferring different contents, such as bioactive lipids, non-coding RNAs, mRNAs, and proteins, and the process is closely related to the occurrence and development of various diseases (*Kalluri and LeBleu, 2020*; *Korvenlaita et al., 2023*; *Fabbiano et al., 2020*). Current research on the biological function of EVs is mostly focused on EVs in body fluids, such as plasma and saliva; however, although their acquisition is relatively simple, such EVs have low tissue specificity (*Yoshimura et al., 2016*; *Li et al., 2017*). In contrast, more in-depth studies have obtained and analyzed EVs from the extracellular environment in organs with complex structures, thus improving tissue specificity (*Mathieu et al., 2019*; *Zhou et al., 2023*; *D'Acunzo et al., 2022*). Nevertheless, this still fails to identify the cell type from which they are derived, making it impossible to carry out more detailed analyses of directional interactions between cells. The development of single-cell/nuclear transcriptome sequencing (snRNA-seq) technology allows us to explore the heterogeneity of cells in complex organs, and the combination of snRNA-seq and tissue EVs transcriptome sequencing (EVs-RNA-seq) can help us clarify the source of tissue EVs and analyze the complex physiological and pathological conditions of organs, using the dual advantages of intracellular and extracellular approaches (*Svensson et al., 2018*; *Stuart et al., 2019*).

Hypoxia is a common stress response in the body that can mediate a series of physiological reactions, including embryonic development and bone marrow hematopoiesis. Hypoxia is also an important factor in various pathologies (*Xiao et al., 2013*; *Wang et al., 2022*; *Choudhry and Harris, 2018*). As the most $O_2$-dependent organ, the brain accounts for 20% of the resting $O_2$ consumption in humans and is thus highly sensitive to hypoxic stress (*Burtscher et al., 2021a*; *Kann and Kovács, 2007*; *Silver and Erecińska, 1998*). Hypoxia is a key cause of many neurological diseases, including Alzheimer's disease, Parkinson's disease (PD), and other age-related neurodegenerative diseases (*Wang et al., 2021*; *Mitroshina et al., 2021*; *Burtscher et al., 2021b*). Although there is known to be a close relationship between hypoxic stress and neurodegeneration, the responses of and damage to brain cells in hypoxic environments remain unclear. The application of snRNA-seq combined with tissue EVs-RNA-seq analysis can be used to comprehensively analyze the interactions among various cell types in the brain under hypoxic stress, thus helping us understand the mechanisms of neurodegenerative injury and identify potential therapeutic molecules for interventions.

Hemoglobin is a major protein involved in the transport and utilization of $O_2$ in the body. It is composed of globin and heme and is mainly expressed in red blood cells (*Gell, 2018*). Previous studies found that hemoglobin also existed in the brain, mainly in neurons, where it was involved in the regulation of oxygen homeostasis and the maintenance of mitochondrial homeostasis, and its expression level was significantly increased under hypoxic stimulation (*Auvinen et al., 2021*; *Singhal et al., 2018*; *Schelshorn et al., 2009*; *Ferrer et al., 2011*). However, recent studies found that hemoglobin may also be involved in the development of neurodegeneration; for example, α-synuclein pathology-hemoglobin-mitochondrial aggregates were found in Lewy bodies in patients with PD (*Killinger et al., 2022*). In addition, neuronal hemoglobin induced dopaminergic neuron loss and cognitive deficits in the substantia nigra in mice (*Santulli et al., 2022*), and hemoglobin deposited in the cerebral cortex after intracerebral hemorrhage exerted neurotoxicity and induced neuronal death in rats (*Wang et al., 2002*). In summary, the specific role of hemoglobin expression in neurons is controversial, and the role of hypoxia in the function of hemoglobin in neurons is still unknown.

In this study, we established a chronic hypoxia mouse model and identified one of the brain's responses to hypoxia. Specifically, we found that non-neuronal cells in the brain regulate neuronal hemoglobin levels via EV-communication under hypoxic conditions, as revealed by the combined analyses of snRNA-seq and EVs-RNA-seq. We also established an in vitro hypoxia neuron model to investigate the specific molecular mechanism of neuroprotection, demonstrating that exosomal hemoglobin helped neurons resist hypoxic damage by maintaining mitochondrial homeostasis. In summary, this study found a new mechanism of exosomal hemoglobin delivery in the brain's hypoxic response 'mutual aid', and clarified the important role of exosomal communication in the brain's hypoxic stress response and resistance to injury. It provides a new research perspective for the prevention and treatment of hypoxic-ischemic-related brain diseases.

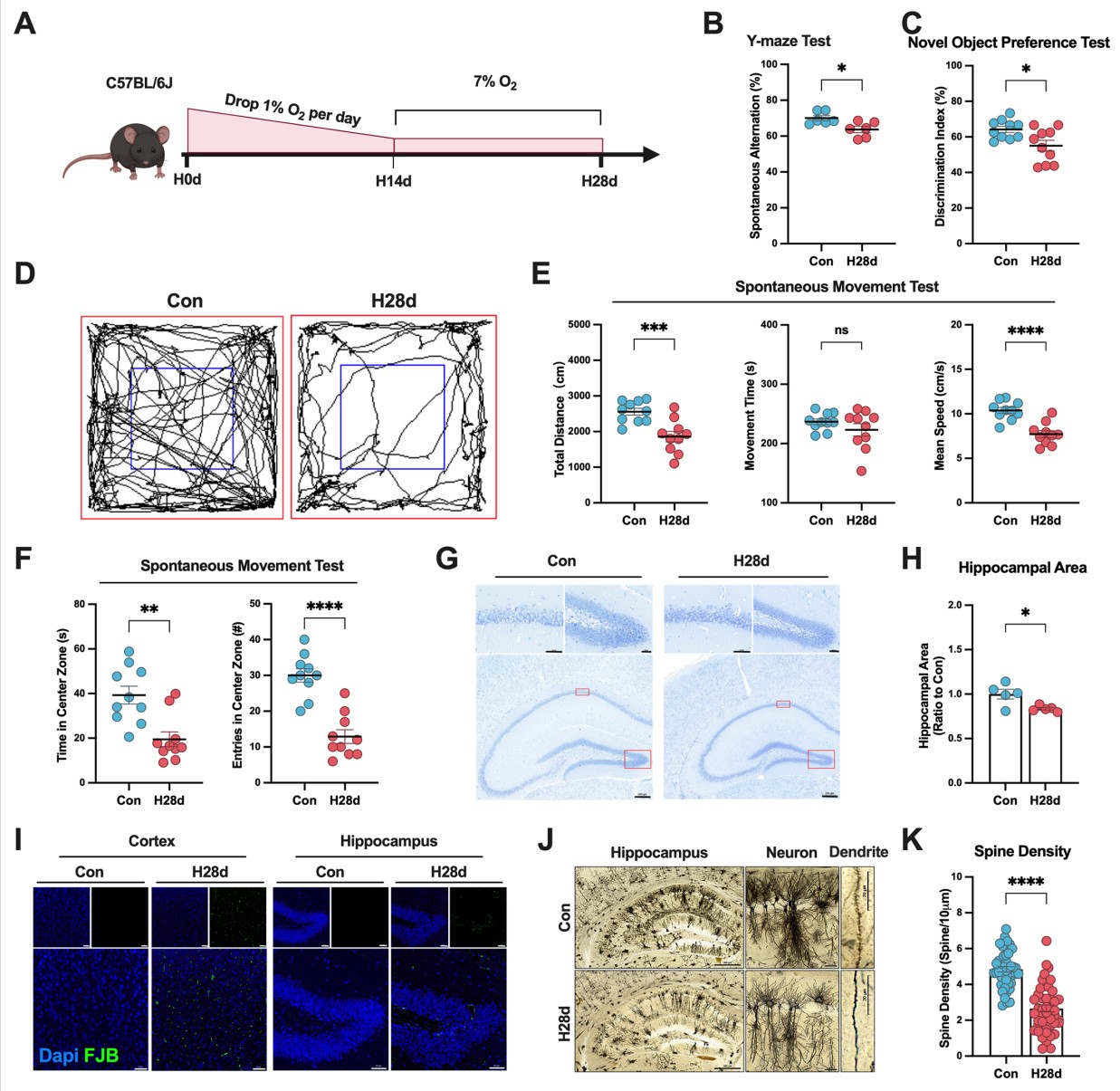

**Figure 1.** Hypoxia induced neurological dysfunction and nerve injury in mice. (**A**) Schematic diagram of chronic hypoxic mouse model construction. Recognition coefficient in Con and H28d mice in (**B**) Y-maze test (n=6) and (**C**) new object recognition test (n=10). (**D**) Trajectories, (**E**) movement index statistics, including movement distance, speed, and time (n=10), and (**F**) cognitive indicators, including length of exploration and number of crossings in the central area (n=10) for Con and H28d mice in open-field tests. (**G**) Nissl staining showing arrangement of neurons in the hippocampus in Con and H28d mice (scale bar: 200 μm in main image, 50 μm in zoomed-in region). (**H**) The area of the hippocampus was counted in each group (n=5). (**I**) Fluoro-Jade B (FJB) staining showing neurodegeneration in the cerebral cortex and hippocampus in Con and H28d mice (scale bar = 50 μm). (**J**) Golgi staining showing dendritic spines of neurons in the hippocampus in Con and H28d mice (scale bars: 500 μm for the hippocampal overview image, 50 μm for the representative neuron, and 20 μm for the dendritic segment). (**K**) The number of dendritic spines on a single axon was counted (n=5). In (**E**) and (**K**), data expressed as mean ± SEM (Welch's *t*-test); other data expressed as mean ± SEM (unpaired *t*-test). ns: not significant, *p<0.05, **p<0.01, ***p<0.001, ****p<0.0001. Con: control group; H28d: mice treated with hypoxia for 28 days.

## Results

### Hypoxia induced the cognitive and motor dysfunction and neuropathological damage

In order to clarify the damage caused by hypoxia in the brain, we established a chronic hypoxia model in C57BL/6J mice using a closed and controllable hypoxic box; the mice were divided into a control

group (Con) and a hypoxic treatment group (hypoxia for 28 days, H28d) (*Figure 1A*). Behavioral tests and neuropathological staining were used to examine the effect of chronic hypoxia on neurological impairment and nerve damage. The results of the Y-maze tests showed that the alternation behavior was significantly decreased, and the novel object recognition tests showed that the exploration time of new objects was significantly reduced after hypoxia, suggesting that hypoxia induced cognitive dysfunction in mice (*Figure 1B and C*). To evaluate spontaneous movement, we performed an open-field test and recorded the motor function and cognitive function indicators of the two groups. The movement trajectory in the open field was significantly reduced in H28d group (*Figure 1D*). Assuming that the total exercise times were similar in both groups, the exercise distance and speed were also significantly reduced in H28d mice (*Figure 1E*). These results suggested that the motor function was impaired after hypoxia. In addition, H28d mice spent less time exploring the central area of the open field and crossing the central area, indicating anxiety-like behavior (*Figure 1F*).

We subsequently examined the brains in the two groups of mice by neuropathological staining to determine if this chronic hypoxic treatment induced nerve damage. Nissl staining showed that neurons in the hippocampal CA1 region were disordered in H28d group, with gaps, abnormal arrangement, and darker cytoplasm compared with Con group, suggesting that the hippocampus was damaged in H28d group (*Figure 1G*). It was also found that the hippocampal region of H28d group showed obvious atrophy (*Figure 1H*). Neuronal degeneration staining (Fluoro-Jade B [FJB] staining) showed neurodegeneration in the cerebral cortex and hippocampal region in mice in H28d group (*Figure 1I*). We also carried out Golgi staining to reveal the morphology and structure of neurons in the brains in the two groups of mice and showed that the density of dendritic spines on neurons was significantly lower in H28d group (*Figure 1J and K*). Above all, chronic hypoxia induced cognitive and motor dysfunction and nerve damage in mice.

## Hypoxia induced the transcription of hemoglobin in non-neuronal cells

To explore the molecular mechanism responsible for chronic hypoxia-induced nerve injury in the brain, we performed snRNA-seq of two groups and visualized the distribution of gene expression in all samples using UMAP visualization and identified 13 cell types according to the expression of cell-specific markers (*Figure 2A*, *Figure 2—figure supplement 1A and B*). We described the proportions of cells before and after hypoxia (*Figure 2B*). Differentially up- and downregulated genes in different cell types were analyzed by gene ontology (GO) enrichment analysis (*Figure 2—figure supplement 1C–L*). Among these, the number of differentially expressed genes (DEGs) in endothelial cells (ECs) was the highest after hypoxia, and most of them were upregulated (*Figure 2C*). We subsequently displayed DEGs in all cell subsets, labeled the top 10 genes, and found the mRNA expression levels of hemoglobin subunits (*Hba-a1*, *Hba-a2*, *Hbb-bs*, *Hbb-bt*) were significantly upregulated in most non-neuronal cells after hypoxia, indicating that might be a common mechanism whereby various non-neuronal cells in the brain respond to hypoxic stress (*Figure 2D*).

We further clarified the interactions among various cell types in the brain under chronic hypoxic stress using the circle diagram and heat map to show the changes in communication among different cell types in the H28d group compared with the Con group (*Figure 2E and F*). We found that the number and intensity of intercellular communications in the brain both increased after hypoxia. Above all, these results suggested that various non-neuronal cells in the brain jointly transcribed hemoglobin under chronic hypoxic stress, which might represent a common mechanism in response to hypoxic stress.

## Hypoxia induced the release of non-neuronal EVs carrying hemoglobin mRNA

SnRNA-seq analysis showed that chronic hypoxia increased intercellular communication in the brain; however, the effects of hypoxia on EVs-mediated communication remain unclear. Hypoxia has been shown to be an important factor inducing EVs release by cells, suggesting that the EVs-mediated communication pathway may be an important mechanism for cells to respond to hypoxia (*Wang et al., 2023*). We therefore performed EVs-RNA-seq of two groups to analyze the mechanisms involved in the brain's resistance to hypoxic damage from the perspective of EVs regulation (*Figure 3A*). We identified EVs isolated from mouse brain tissue. Transmission electron microscopy confirmed that the diameter of the EVs was approximately 100 nm and displayed a characteristic cup-shaped morphology

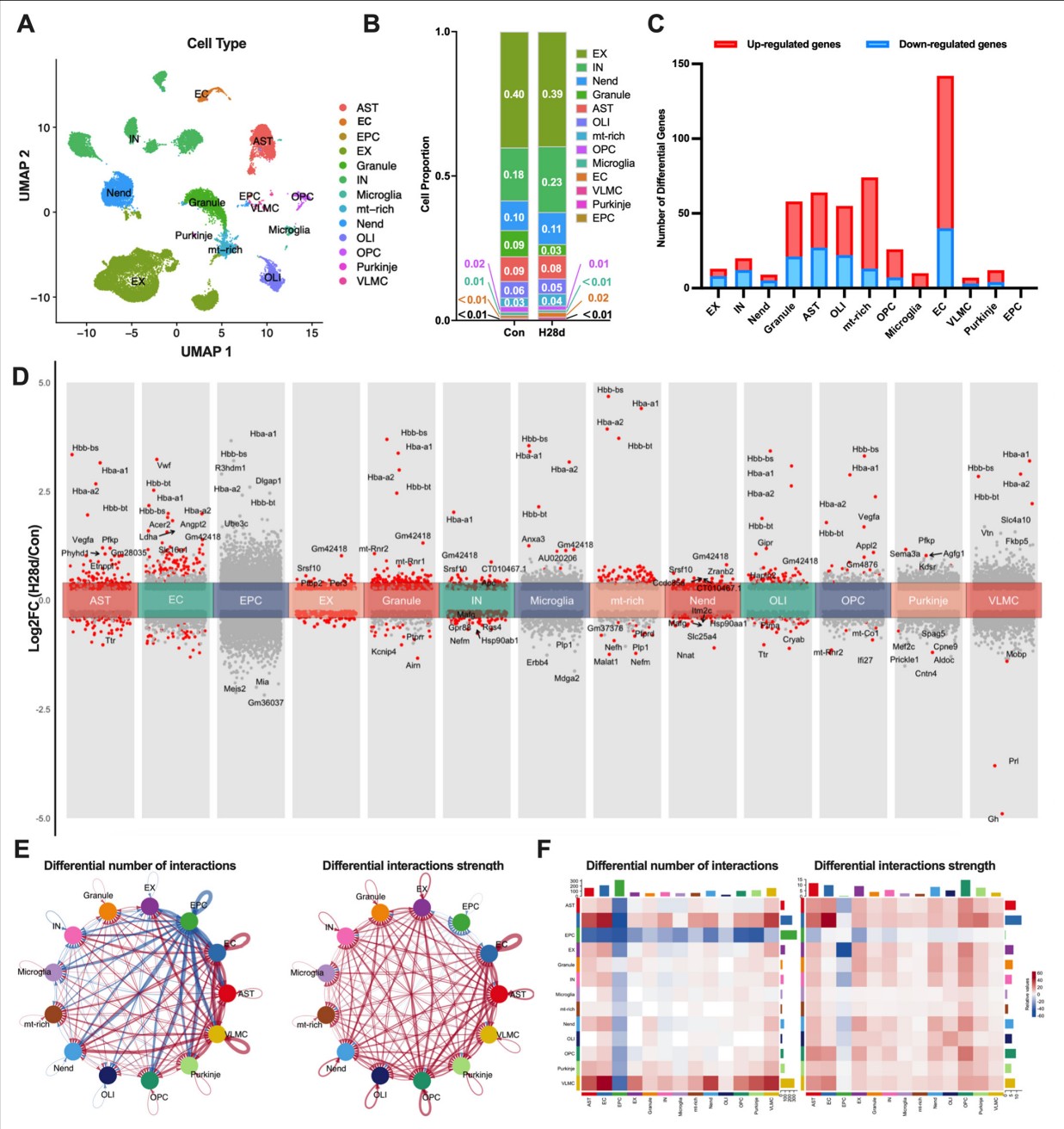

**Figure 2.** Chronic hypoxia increased hemoglobin transcription in mouse brain. (**A**) Transcription profiles of cell subsets in the brain in Con and H28d mice visualized by UMAP. (**B**) Proportions of different cells in the brain in Con and H28d mice. (**C**) Histogram showing numbers of differentially expressed genes (DEGs) in different cell types in mouse brain after hypoxia. Up- and downregulated forms displayed separately. (**D**) DEGs in different cell types in mouse brain after hypoxia shown in stacked volcano plots. Top 10 DEGs marked based on absolute log2FC value. (**E**) Circle diagram showing number and intensity of intercellular communications among different cell types in the brain in Con and H28d mice. Red indicates increase and blue indicates decrease. (**F**) Communications among cells in the H28d and Con groups demonstrated by combination of heat map and histogram. Con: control group; H28d: mice treated with hypoxia for 28 days. AST: astrocyte; EC: endothelial cell; EPC: ependymocyte; EX: excitatory neuron; Granule: granule cell; IN: inhibitory neuron; Microgila: microgila; mt-rich: mitochondrial-rich cell; Nend: neuronendocrine cell; OLI: oligodendrocyte; OPC: oligodendrocyte precursor cell; Purkinje: Purkinje cell; VLMC: vascular lepotomeningeal cell.

The online version of this article includes the following figure supplement(s) for figure 2:

**Figure supplement 1.** SnRNA-seq analysis identified 13 cell types in the mouse brain.

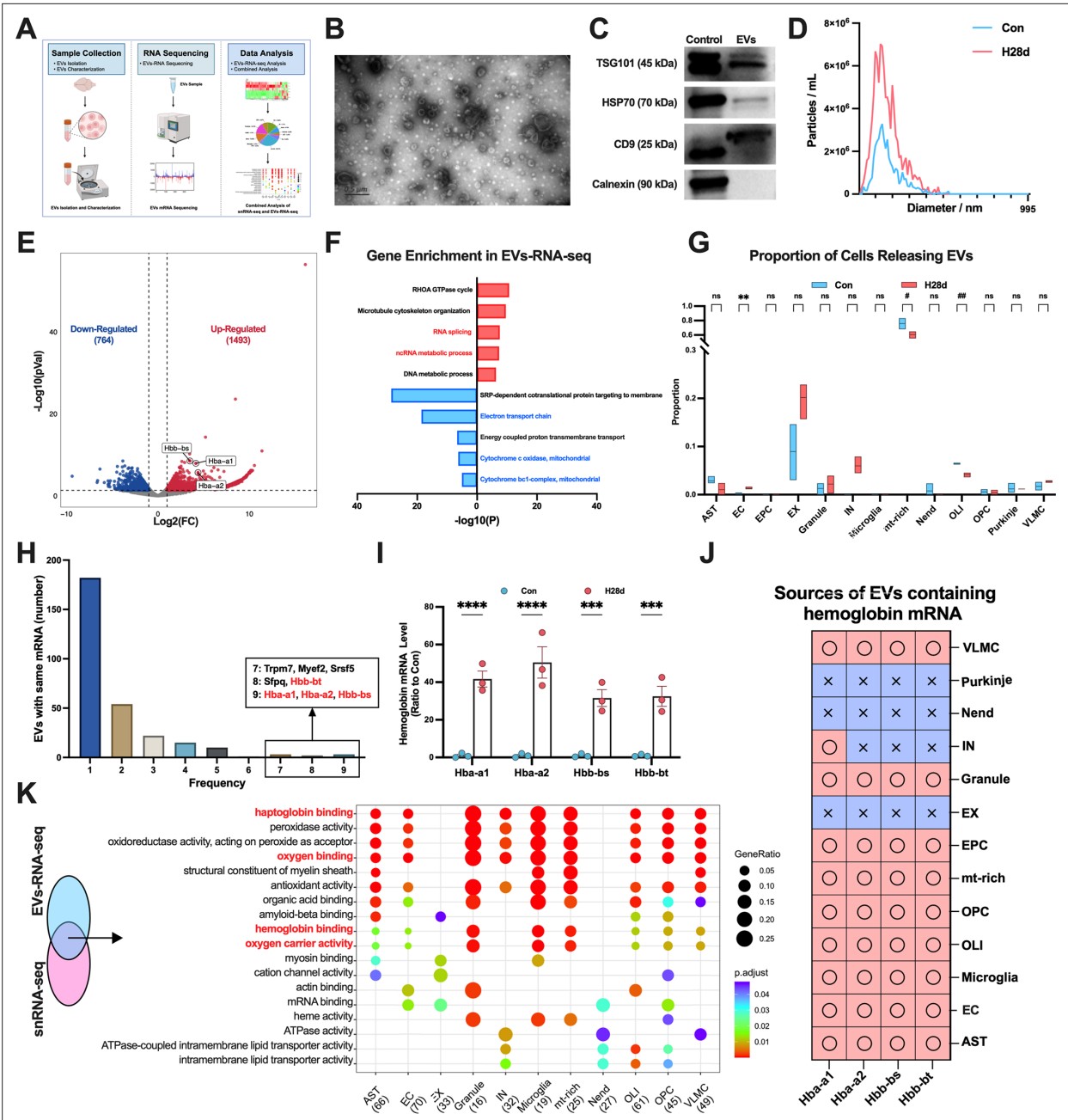

**Figure 3.** Hypoxia increased release of extracellular vesicles (EVs) carrying hemoglobin mRNA in mouse brain. (**A**) Schematic diagram of EV extraction and transcriptome sequencing in brain tissue from Con and H28d mice. (**B**) Typical electron microscope images of brain tissue EVs (scale bar = 0.5 μm). (**C**) Western blot analysis of EVs markers TSG101, HSP70, and CD9 and EVs negative marker calnexin in EVs isolated from cell lysates and brain tissues. (**D**) Nanoparticle tracking analysis (NTA) of brain tissue EVs isolated from Con and H28d mice. (**E**) Differentially expressed genes (DEGs) carried by mouse brain tissue EVs after hypoxia displayed by volcanic maps. Hemoglobin subunits labeled. (**F**) Histogram showing enriched signaling pathways for top five up- and downregulated DEGs in EVs-RNA-seq. (**G**) The proportion of cells releasing EVs in each group. Data expressed as mean ± SEM (two-way ANOVA). *Upregulated, #Downregulated. ns: not significant, **p<0.01, #p<0.05, ##p<0.01. (**H**) Histogram illustrating the frequencies of cell types secreting EVs containing the same DEG mRNAs. The mRNAs co-carried by EVs secreted by seven, eight, and nine cells are displayed. (**I**) mRNA levels of each hemoglobin subunit in brain tissue EVs from Con and H28d mice (n=3). Data expressed as mean ± SEM (two-way ANOVA). ***p<0.001, ****p<0.0001. (**J**) Heat map showing cell sources of EVs carrying hemoglobin subunit mRNAs. (**K**) GO enrichment analysis of DEGs obtained by combined snRNA-seq and EVs-RNA-seq. Signal pathways of more than three cell types displayed in a bubble map. Con: control group; H28d: mice treated with hypoxia for 28 days. AST: astrocyte; EC: endothelial cell; EPC: ependymocyte; EX: excitatory neuron; Granule: granule cell; IN: inhibitory neuron; Microgila: microgila; mt-rich: mitochondrial-rich cell; Nend: neuronendocrine cell; OLI: oligodendrocyte; OPC: oligodendrocyte precursor cell; Purkinje: Purkinje cell; VLMC: vascular lepotomeningeal cell.

*Figure 3 continued on next page*

*Figure 3 continued*

The online version of this article includes the following source data and figure supplement(s) for figure 3:

**Source data 1.** Uncropped blots for *Figure 3C* with relevant bands marked.

**Source data 2.** Original uncropped image files for western blot in *Figure 3C*.

**Figure supplement 1.** EVs-RNA-seq analysis showed that mitochondrial function was impaired in the mouse brain after hypoxia.

**Figure supplement 2.** mRNA in endothelial cell-derived extracellular vesicles (EVs).

(*Figure 3B*). Western blot analysis revealed positive expression of EV markers TSG101, HSP90, and CD9, while calnexin was not detected (*Figure 3C*). Nanoparticle tracking analysis indicated that the predominant size of the isolated EVs ranged from 50 to 200 nm, with increased EV secretion observed in the brains following hypoxia (*Figure 3D*).

After the transcriptome sequencing of EVs isolated from Con and H28d mice, we analyzed the DEGs between these two groups (*Figure 3—figure supplement 1A and B*). A total of 2257 DEGs were identified, of which 1493 were upregulated and 764 were downregulated after hypoxia. Hemoglobin mRNA was also significantly upregulated in EVs after hypoxia, consistent with the snRNA-seq results (*Figure 3E*). We conducted gene enrichment analysis on all identified up- and downregulated DEGs and selected the top 5 enriched pathways for presentation. We found that several pathways related to RNA metabolism were significantly upregulated, while several pathways associated with energy metabolism were downregulated (*Figure 3—figure supplement 1C–F*, *Figure 3F*). These results suggested that energy metabolism pathways in the brain may have been damaged by hypoxia, and various cell types in the brain may regulate RNA metabolism via the EVs pathway to resist hypoxic injury. The protein–protein interaction (PPI) analysis yielded results consistent with the aforementioned analysis (*Figure 3—figure supplement 1C–H*). The Gene Set Enrichment Analysis (GSEA) showed that the function of mitochondria in the brain was also severely impaired after hypoxia (*Figure 3—figure supplement 1I and J*).

We compared the DEGs obtained from EVs-RNA-seq with the snRNA-seq and screened out DEGs that changed consistently within the cell and in the extracellular environment after hypoxia. Our findings revealed the proportion of cells releasing EVs in each group, indicating a significant increase in EVs secretion solely by brain EC following hypoxia (*Figure 3G*, *Figure 3—figure supplement 2A and B*). Multiple differential mRNAs were present in EVs secreted by various types of cells after hypoxia, including mRNAs of hemoglobin subunits, suggesting that this may be a common way for multiple cells to respond to hypoxia via the EVs pathway (*Figure 3H*). We therefore analyzed the hemoglobin mRNA levels in EVs from the two groups, and found that exosomal mRNA levels of hemoglobin were significantly increased after hypoxia, but only existed in EVs secreted by non-neuronal cells (*Figure 3I and J*). We subsequently performed GO analysis on all DEGs with consistent changes in both snRNA-seq and EVs-RNA-seq according to cell types for GO analysis and presented the common signaling pathways accumulated in three or more cell types. Many of these pathways were related to hemoglobin function, suggesting that changes in hemoglobin after hypoxia induction could be a common means by which various types of cells respond to hypoxia (*Figure 3K*). Above all, chronic hypoxia can enhance EVs communication in the brain and induce non-neuronal cells to transcribe hemoglobin and release them in the form of EVs.

## Hypoxia induced the expression of hemoglobin in neurons

Combined analysis of snRNA-seq and EVs-RNA-seq showed numerous hemoglobin mRNA existed in the intracellular and extracellular environments in the brain after hypoxia, which may provide a common mechanism for various cell types to cope with hypoxic injury. However, changes in hemoglobin protein levels after hypoxia were still unclear. We observed brain tissues from two groups of mice and found that brain tissue in the H28d group remained red after perfusion with normal saline, indicating large amounts of hemoglobin in the brain (*Figure 4A*). We further explored the transcriptional changes in hemoglobin subunits in cells after hypoxia by snRNA-seq and visualized the results in a violin plot, and the mRNA levels of hemoglobin subunits were low in all cell types in the Con group, but were increased in non-neuronal cells after hypoxia (*Figure 4B*). RT-qPCR confirmed that *Hba-al* mRNA levels in the brain were significantly increased after hypoxia (*Figure 4C*). Western blotting and

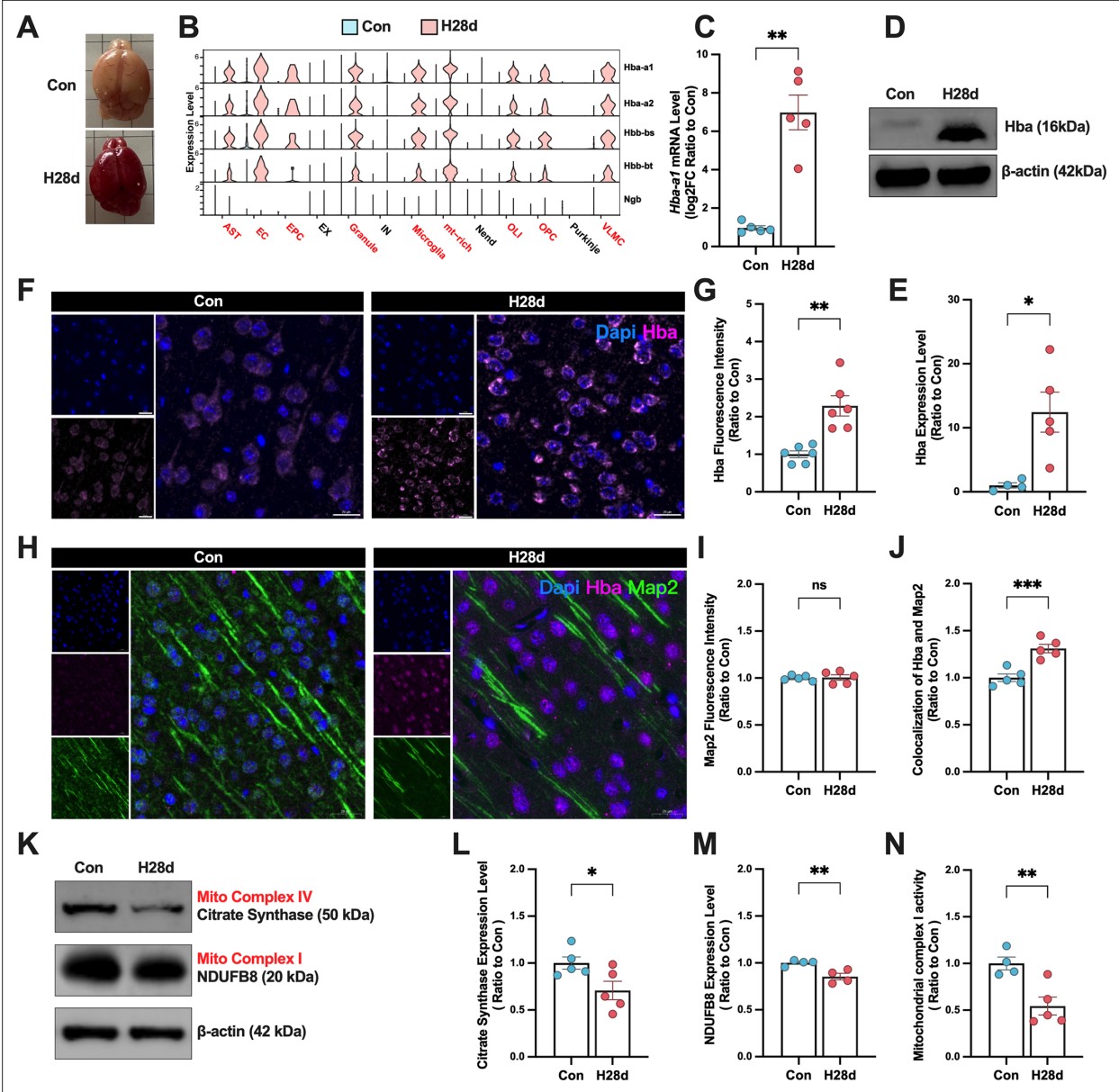

**Figure 4.** Hypoxia increased hemoglobin expression in mouse neurons. (**A**) Brains of Con and H28d mice. (**B**) Expression levels of hemoglobin subunits in snRNA-seq in different cell types in the brain shown by violin maps. (**C**) *Hba-a1* mRNA levels in brain in Con and H28d mice detected by real-time qPCR (n=5). (**D, E**) Hba expression levels in the brain in Con and H28d mice detected by western blotting (n=5). (**F, G**) Hba expression levels in the brain in Con and H28d mice detected by immunofluorescence (scale bar = 20 µm, n=6). (**H**) Expression levels of Hba and Map2 in the brain in Con and H28d mice detected by immunofluorescence (scale bar = 20 µm, n=5). (**I**) Fluorescence intensity of Map2 in the brain in Con and H28d mice. (**J**) Co-localization of Hba and Map2 in the brain in Con and H28d mice. (**K–M**) Expression levels of mitochondrial complex I marker NDUFB8 and mitochondrial complex IV marker citrate synthase in the brain in Con and H28d mice detected by western blotting (n=4–5). (**N**) Activity of mitochondrial complex I in the brain in Con and H28d mice (n=4–5). In (**C**), (**E**), and (**G**), data expressed as mean ± SEM (Welch's *t*-test); other data expressed as mean ± SEM (unpaired *t*-test). ns: not significant, *p<0.05, **p<0.01, ***p<0.001. Con: control group; H28d: mice treated with hypoxia for 28 days.

The online version of this article includes the following source data for figure 4:

**Source data 1.** Uncropped blots for *Figure 4D and K* with relevant bands marked.

**Source data 2.** Original uncropped image files for western blot in *Figure 4D and K*.

immunofluorescence also revealed that Hba protein levels were also significantly increased in the H28d group (*Figure 4D–G*).

Although previous studies showed that transient hypoxia induced the expression of hemoglobin in neurons in the brain, its specific function is still unclear (*Schelshorn et al., 2009*). We therefore speculated that hemoglobin expression levels may also be increased in neurons after hypoxia. We labeled Hba and Map2 by immunofluorescence and found that the Hba fluorescence intensity in Map2-positive cells increased significantly after hypoxia, indicating that hypoxia induced hemoglobin specifically expressed and accumulated in neurons (*Figure 4H–J*). SnRNA-seq and EVs-RNA-seq showed that chronic hypoxia-induced hemoglobin transcription was mainly increased in non-neuronal cells and their secreted EVs (*Figure 3J* and *Figure 4B*), but immunofluorescence showed that hypoxia-induced increased hemoglobin was specifically present in neurons, confirming that non-neuronal cells in the brain could dynamically regulate hemoglobin levels in neurons via the EVs pathway under hypoxia, thus helping neurons to resist the hypoxic damage.

Neuroglobin (Ngb) is a recently discovered type of oxygen-carrying globin, which under hypoxic conditions can facilitate the diffusion of oxygen into mitochondria, enhancing tissue oxygen utilization. Particularly, it plays a crucial protective role in neural tissues during cerebral ischemia and hypoxia. To elucidate whether Ngb exerts protective effects in the brains of H28d mice, we conducted an analysis of snRNA-seq data. Visualization of Ngb transcription levels revealed consistently low expression levels across various brain cell types before and after hypoxia, contrasting with the behavior of hemoglobin. Consequently, this study continued its focus on elucidating the precise molecular mechanisms underlying hemoglobin's response to hypoxia.

Mitochondria are the main organelles involved in aerobic respiration and are highly sensitive to oxygen concentrations. We therefore determined if hypoxia induced mitochondrial damage in brain. Western blotting showed that mitochondrial complex I and IV were both significantly decreased in the brain after hypoxia, through analyzing the markers of mitochondrial complex I (NDUFB8) and complex IV (citrate synthase) (*Figure 4K–M*). We further detected the activity of mitochondrial complex I in the brain in both groups and showed that its activity was significantly decreased after hypoxia (*Figure 4N*). These results confirm that chronic hypoxia can damage mitochondrial activity in the brain. Above all, these results indicate that hypoxia-induced non-neuronal transcriptionally increased hemoglobin is transmitted to neurons via the EVs pathway.

## Hypoxia specifically increased hemoglobin transcription in endothelial cells in vitro

We constructed an in vitro hypoxia model to clarify the specific protective mechanism of exosomal hemoglobin produced in hypoxia-induced neurons on mitochondria. In view of the most significant increase in cell communication intensity and the highest transcription of hemoglobin in ECs after hypoxia, we used the human brain microvascular endothelial cells (hcMEC/D3 cells) as a representative of non-neuronal cells in the brain, and the human neuroblastoma cells (SH-SY5Y cells) as a cell model of hypoxia-induced nerve injury. We established four different hypoxia-duration groups using an oxygen concentration of 1% and detected the effects on cell viability and cytotoxicity. The viability of hcMEC/D3 cells was significantly reduced in the H4h and subsequent groups, compared with the H0h group, and the cytotoxicity was significantly increased in the H12h group (*Figure 5A and B*). The viability of SH-SY5Y cells was significantly reduced in the H4h and subsequent groups, compared with the H0h group, and the cytotoxicity was significantly increased in the H4h group and subsequent groups as well (*Figure 5C and D*).

Considering the mitochondrial damage in the mouse hypoxia models, we detected the mitochondrial status of SH-SY5Y cells before and after hypoxia. We labeled the mitochondria using MitoTracker and detected the fluorescence intensity in each group of SH-SY5Y cells by flow cytometry to determine the number of mitochondria (*Figure 5E*). Fluorescence intensity in SH-SY5Y cells decreased gradually with hypoxia time, indicating that the number of mitochondria was significantly reduced by prolongation of hypoxia (*Figure 5F*). The mitochondrial membrane potential is an important indicator of mitochondrial function. Its reduction will induce mitochondrial homeostasis damage in cells, which in turn induces cell damage. We therefore evaluated the effects of hypoxia by JC-1 staining and found the mitochondrial membrane potential also significantly decreased with prolongation of hypoxia (*Figure 5G*). These results indicate that severe mitochondrial damage

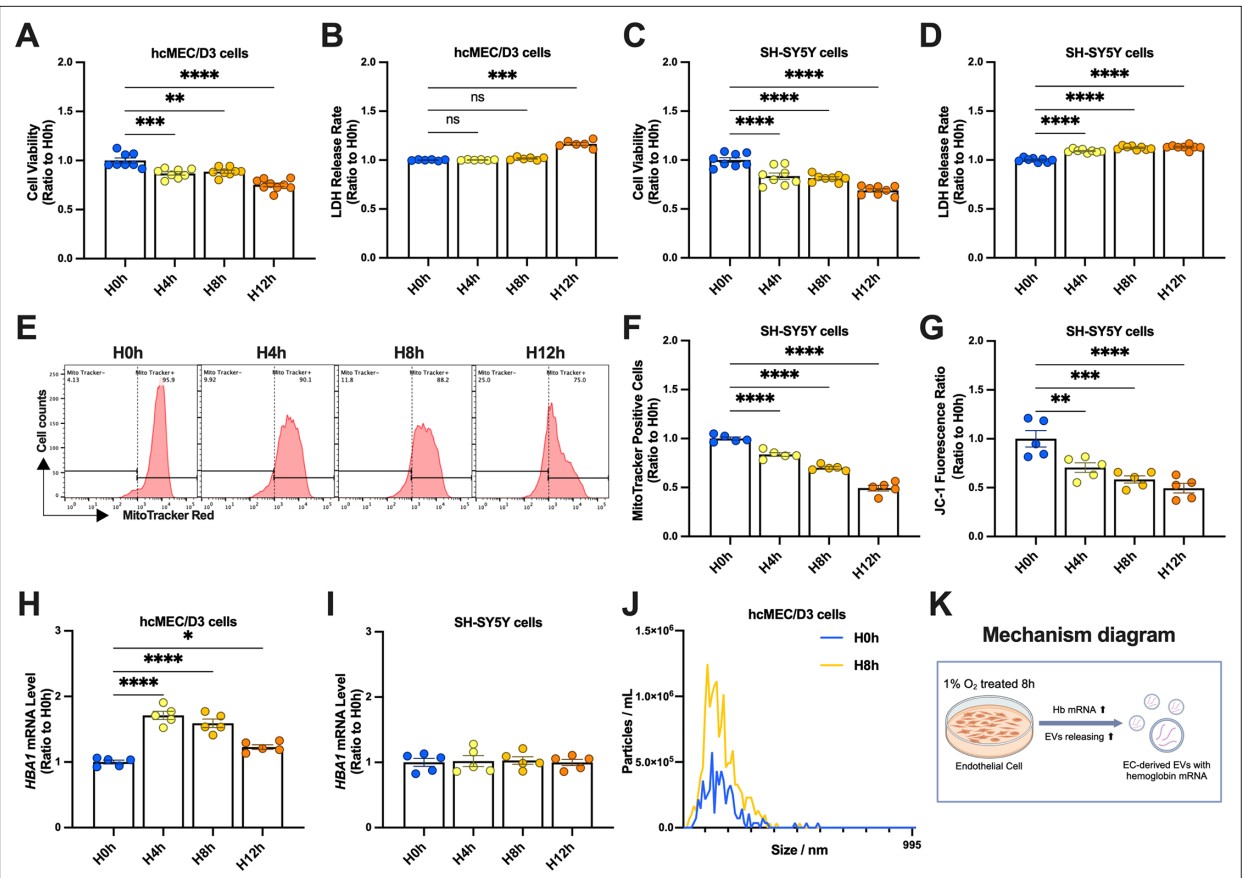

**Figure 5.** Hypoxia increased hemoglobin transcription in endothelial cells and induced release of extracellular vesicles (EVs). (**A**) Viability of hcMEC/D3 cells detected by CCK-8 assay in H0h, H4h, H8h, and H12h groups (n=8) and (**B**) cytotoxicity in each group detected by lactate dehydrogenase (LDH) assay (n=6). (**C**) Viability of SH-SY5Y cells in each group detected by CCK-8 assay (n=8) and (**D**) cytotoxicity in each group detected by LDH assay (n=8). (**E, F**) Mitochondria were labeled in SH-SY5Y groups using MitoTracker staining, and the number of mitochondria in each group was analyzed and counted by flow cytometry (n=5). (**G**) Mitochondrial membrane potential in each SH-SY5Y group was detected by JC-1 labeling and analyzed by flow cytometry (n=5). *HBA1* mRNA levels in (**H**) hcMEC/D3 cells and (**I**) SH-SY5Y cells detected by real-time qPCR (n=5 each). (**J**) Nanoparticle tracking analysis (NTA) of EVs in hcMEC/D3 cells treated with hypoxia for 0 and 8 h. (**K**) Mechanism diagram. In (**B**), data expressed as mean ± SEM (Brown–Forsythe ANOVA); other data as mean ± SEM (one-way ANOVA). ns: not significant, *p<0.05, **p<0.01, ***p<0.001, ****p<0.0001. H0, 4, 8, 12 h: cells treated with hypoxia for 0, 4, 8, 12 h.

The online version of this article includes the following figure supplement(s) for figure 5:

**Figure supplement 1.** H8h treatment induces the most of extracellular vesicles (EVs) release from hcMEC/D3 cells.

occurred when neurons encountered hypoxic stress, which may be a key factor in hypoxic nerve injury.

To further simulate the response of non-neuronal cells to transcribed hemoglobin and EVs to neurons and express in the former research, we first detected *HBA1* mRNA levels in hcMEC/D3 and SH-SY5Y cells in each group. The level of *HBA1* mRNA was significantly upregulated in hcMEC/D3 cells in the H4h group, and then slowly decreased to pre-hypoxia levels (*Figure 5H*). However, level of *HBA1* mRNA in SH-SY5Y cells was unaffected by hypoxia (*Figure 5I*). We then isolated and analyzed EVs from hcMEC/D3 cells under different treatment durations and showed that EVs were significantly increased in the H8h group compared with the H0h group, indicating that hypoxia for 8 h induced EVs release by hcMEC/D3 cells (*Figure 5J*, *Figure 5—figure supplement 1A–C*).

Above all, we established an experimental hypoxia method for EC and neurons in vitro, and found that hypoxia treatment for 8 h induced a significant transcription of hemoglobin in EC, which was subsequently released into the culture medium via EVs (*Figure 5K*).

# Neurons resisted hypoxic injury by expressing exosomal hemoglobin in vitro

Based on the results showing that hypoxia treatment for 8 h had no significant toxic effect on hcMEC/D3 cells, and hcMEC/D3 cells showed the most significant increase in *HBA1* transcription and the largest release of EVs, we selected conditioned medium from H8h hcMEC/D3 cells as the donors of hemoglobin mRNA EVs in subsequent in vitro experiments, with conditioned medium from H0h hcMEC/D3 cells as the control. To determine if the EVs secreted by EC under hypoxia were taken up and utilized by neurons, we first isolated EVs from hcMEC/D3 H8h cells by ultracentrifugation and co-cultured them with SH-SY5Y cells after fluorescence labeling. The results showed that hcMEC/D3 H8h EVs were actively taken up by SH-SY5Y cells, thus providing a basis for subsequent exploration of the neuroprotective effect of EC-derived EVs (*Figure 6A*). In order to determine if EVs secreted by EC helped neurons to dynamically regulate hemoglobin levels and resist hypoxia-induced nerve damage, we used EC-conditioned medium exchange experiments. Cells were subjected to hypoxia for 0 (H0h) or 8 h (H8h), hypoxia for 8 h+control medium (H8h+H0h [EC]), and hypoxia for 8 h+conditioned medium (H8h+H8h [EC]). In the H8h+H0h (EC) and H8h+H8h (EC) groups, endothelial cell H0h (+H0h [EC]) and H8h medium (+H8h [EC]) were exchanged into SH-SY5Y cells, followed by hypoxic treatment. We initially demonstrated that EC-conditioned medium does not cause damage to the cell viability and mitochondrial homeostasis of SH-SY5Y cells (*Figure 6—figure supplement 1A–D*).

We detected *HBA1* mRNA levels in SH-SY5Y cells in each group. *HBA1* mRNA levels were significantly upregulated in the H8h+H8h (EC) group, suggesting that conditioned medium from EC treated with hypoxia for 8 h could increase the transcription of hemoglobin after hypoxia (*Figure 6B*). We also detected the protein levels of Hba by immunofluorescence and showed that Hba protein levels were significantly increased in the H8h+H8h (EC) group, confirming that H8h conditioned medium from EC could increase hemoglobin levels in neurons (*Figure 6C and D*).

We confirmed that hypoxia treatment for 8 h decreased SH-SY5Y cell viability, the number of mitochondria, and mitochondrial membrane potential. We therefore detected the cell and mitochondrial viability of SH-SY5Y cells supplemented with EC-derived conditioned medium and hypoxia, according to the above experimental protocol (*Figure 6E*). Treatment with H8h EC-conditioned medium improved the number of mitochondria and mitochondrial membrane potential (*Figure 6F and G*) and improved the viability of neurons after hypoxia for 8 h (*Figure 6H*).

To verified the function of exosomal hemoglobin in helping neurons to resist hypoxic injury in primary neurons (PNs) in vitro, we designed the following experiments. The results of immunofluorescence showed that hcMEC/D3 H8h EVs were actively taken up by PN (*Figure 6I*). The PN were then subjected to EC-conditioned medium exchange combined with hypoxia treatment to determine if conditioned medium could increase the transcription and expression levels of hemoglobin. The mRNA and protein levels of hemoglobin were significantly upregulated in the H8h+H8h (EC) group (*Figure 6J–L*). JC-1 staining showed that the mitochondrial membrane potential was significantly higher in the H8h+H8h (EC) group than in the H8h group (*Figure 6M*). MitoTracker staining showed that mitochondrial numbers were significantly increased in the H8h+H8h (EC) group compared with the H8h group (*Figure 6N*). We also evaluated the viability of PN in each group by PI/Hoechst staining and showed that their cell viability was significantly reduced after 8 h of hypoxia treatment, and that this damage could be attenuated by EC-conditioned medium (*Figure 6O*). Above all, we found that EC-derived EVs carrying hemoglobin mRNA could regulate hemoglobin levels in neurons under hypoxic conditions, aiding neurons in maintaining mitochondrial homeostasis and resisting hypoxic damage (*Figure 6P*).

To confirm that the protective effect of the EC-derived conditioned medium in terms of regulating hemoglobin levels in neurons was derived from EVs, we removed EVs from each group of medium and repeated the above experiments (*Figure 7A*). After removing EVs, we verified that the resulting conditioned medium failed to increase hemoglobin in SH-SY5Y cells (*Figure 6—figure supplement 2*), and the H8h EC-conditioned medium lost its abilities to improve mitochondrial homeostasis and neuronal viability under hypoxic stress (*Figure 7B–D*).

We further confirmed that the protective effect of EC-derived conditioned medium in regulating hemoglobin levels in neurons was derived from *HBA1* mRNA carried by EVs by pretreating the EC with *HBA1* RNAi plasmid interference (*Figure 7E*, *Figure 7—figure supplement 1A–C*). We screened plasmids for specific knock-down of *HBA1* mRNA in hcMEC/D3 cells, and verified that the resulting

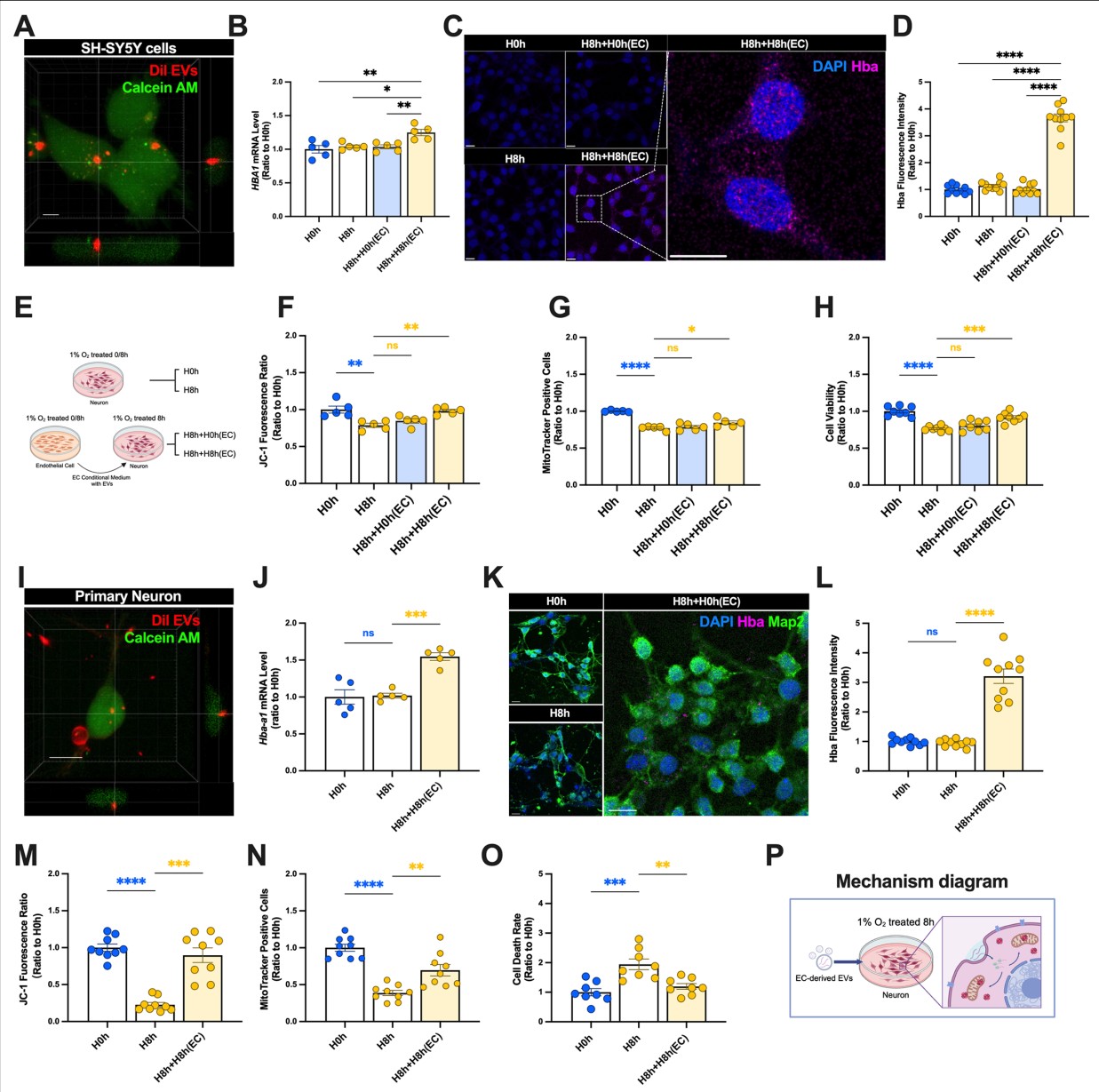

**Figure 6.** Hypoxia induced neuronal uptake of endothelial cell-derived extracellular vesicles (EVs) and expression of hemoglobin. (**A**) EVs in medium from H8h hcMEC/D3 cells were labeled with Dil dye and added to SH-SY5Y cells labeled with Calcein AM for 24 h. Uptake of hcMEC/D3 EVs by SH-SY5Y cells was observed by confocal microscopy (scale bar = 5 μm). (**B**) *HBA1* mRNA levels in SH-SY5Y cells detected by real-time qPCR (n=5). (**C, D**) Hba1 expression levels in SH-SY5Y cells detected by immunofluorescence (scale bar = 10 μm) (n=10). (**E**) hcMEC/D3 conditioned medium treatment diagram. (**F**) Viability of SH-SY5Y cells (n=8–10). (**G**) Numbers of mitochondria in SH-SY5Y cells (n=5). (**H**) Mitochondrial membrane potential in SH-SY5Y cells (n=5). (**I**) EVs in medium from H8h hcMEC/D3 cells were labeled with Dil and then added to calcein AM-labeled primary neurons for 24 h. Uptake of hcMEC/D3 EVs by primary neurons was observed by confocal microscopy (scale bar = 5 μm). (**J**) *Hba-a1* mRNA levels in primary neurons detected by real-time qPCR (n=5). (**K, L**) Hba expression levels in primary neurons detected by immunofluorescence (scale bar = 10 μm) (n=10). (**M**) Viability of primary neurons in each group characterized by PI/Hoechst staining and photographed by confocal microscopy (n=8). (**N**) Number of mitochondria in primary neurons in each group characterized by MitoTracker staining and photographed by confocal microscopy (n=9). (**O**) Mitochondrial membrane potential of primary neurons in each group characterized by JC-1 staining and photographed by confocal microscopy (n=9). (**P**) Mechanism diagram. In (**L**), data expressed as mean ± SEM (Brown–Forsythe ANOVA); other data as mean ± SEM (one-way ANOVA). ns: not significant, *p<0.05, **p<0.01, ***p<0.001, ****p<0.0001.

The online version of this article includes the following figure supplement(s) for figure 6:

**Figure supplement 1.** Endothelial-derived culture media does not cause neuronal damage.

**Figure supplement 2.** Extracellular vesicle (EV) removal treatment disrupts the transfer of hemoglobin mRNA in EC-conditioned medium.

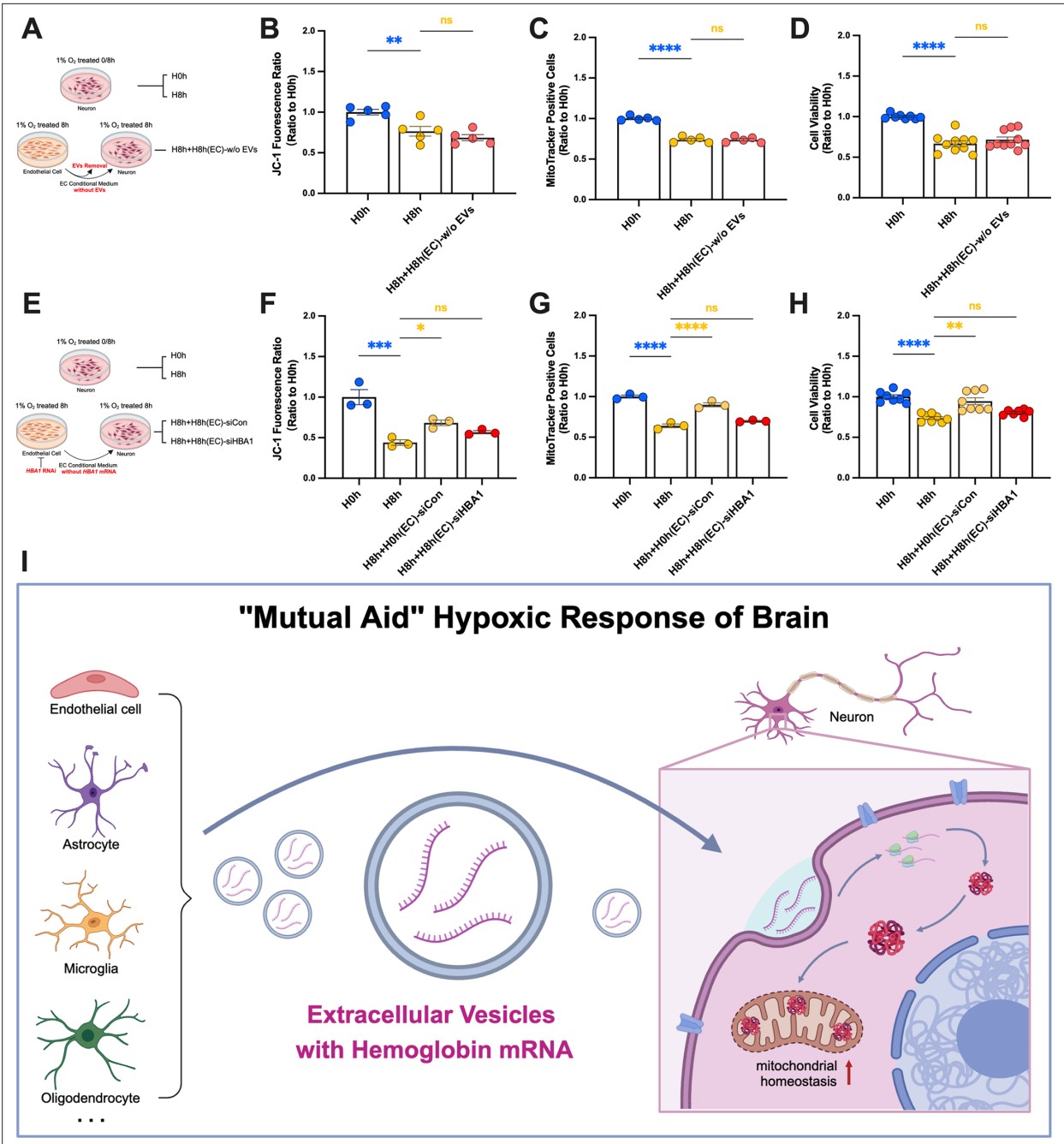

**Figure 7.** Exosomal hemoglobin helped neurons to resist hypoxic injury by maintaining neuronal mitochondrial homeostasis. (**A**) hcMEC/D3 conditioned medium extracellular vesicle (EV) removal pretreatment diagram. (**B–D**) Viability, mitochondrial number, and mitochondrial membrane potential of SH-SY5Y cells (n=5–8). (**E**) HBA RNAi pretreatment diagram of hcMEC/D3 conditioned medium. (**F–H**) Viability, mitochondrial number, and mitochondrial membrane potential of SH-SY5Y cells (n=3–8). (**I**) Mechanism diagram. In (**H**), data expressed as mean ± SEM (Brown–Forsythe ANOVA); other data as mean ± SEM (one-way ANOVA). ns: not significant, *p<0.05, **p<0.01, ***p<0.001, ****p<0.0001.

The online version of this article includes the following figure supplement(s) for figure 7:

**Figure supplement 1.** *HBA1* siRNA treatment disrupts the transfer of hemoglobin mRNA in endothelial cell (EC)-conditioned medium.

conditioned medium failed to increase hemoglobin in SH-SY5Y cells (*Figure 7—figure supplement 1D–F*). Following HBA1 RNAi treatment of EC, the conditioned medium lost its abilities to improve mitochondrial homeostasis and neuronal viability under hypoxic stress (*Figure 7F–H*).

In summary, this study found that EVs containing hemoglobin mRNA secreted by EC after hypoxia treatment regulated the hemoglobin expression in neurons, thus helping neurons to resist mitochondrial dysfunction and nerve injury under hypoxia stress (*Figure 7I*).

## Discussion

The brain is the most vulnerable organ to hypoxic injury and hypoxia is associated with the occurrence and development of various central nervous system diseases; however, the specific molecular mechanisms are still unclear. Beyond the hypoxia-inducible factor signaling pathways, intercellular communication among diverse brain cell types is imperative for mitigating stress-induced damage in hypoxic conditions. Mounting evidence suggests that hypoxia enhances intercellular communication primarily through interactions involving EVs, thereby facilitating the transmission of various cargoes with distinct functional roles (*Korvenlaita et al., 2023*; *Wang et al., 2023*). Our research findings also indicate that hypoxia induces a significant enhancement in EVs-mediated communication among various cell types in the brain. Through the isolation and purification of tissue-derived EVs followed by transcriptomic sequencing analysis, we are able to describe changes in intercellular communication within the brain from an extracellular perspective. However, this method does not provide a comprehensive map of intracellular communication (*Zhou et al., 2023*). Through the combined analysis of snRNA-seq and EVs-RNA-seq, we can accurately identify the cellular origin of brain-derived EVs, thus comprehensively revealing changes in intercellular communication within the brain induced by hypoxia.

To investigate this phenomenon, we employed a chronic hypoxia model in which mice were exposed to 7% $O_2$ for 28 days. This model aims to mimic systemic hypoxia-induced multi-organ damage, a condition observed in diseases such as high-altitude hypoxia, chronic obstructive pulmonary disease, and acute hypoxic brain injury. The primary goal of this model is to explore how the brain adapts under extreme low-oxygen conditions and employs specific mechanisms to protect itself from hypoxia-induced damage. This approach provides valuable insight into diseases related to hypoxic-ischemic injury in the brain, including stroke and vascular dementia, offering a novel perspective for potential clinical applications.

This study elucidates a novel mechanism whereby various non-neuronal cells in the brain regulate hemoglobin levels in neurons through EV-communication under hypoxic stress, emphasizing the crucial role of EV communication in the brain's response to hypoxia.

Hemoglobin, as the main protein carrying and transporting oxygen molecules in the body, is also expressed in neurons, but its precise function remains controversial (*Schelshorn et al., 2009*; *Russo et al., 2013*; *Zheng et al., 2022*). Current research suggests that hemoglobin expressed in the brains of PD patients participates in regulating intraneuronal oxygen homeostasis and maintaining mitochondrial stability (*Vanni et al., 2018*). However, other studies indicate that hemoglobin may exert neurotoxic effects by inducing oxidative stress and neuronal death, thereby contributing to the onset and progression of neurodegeneration (*Santulli et al., 2022*; *Wang et al., 2002*). To clarify the role of hemoglobin in the brain under hypoxic stress, this study, based on the elucidation of the mechanism by which non-neuronal cells regulate neuronal hemoglobin through EVs, conducted in vitro experiments. These experiments revealed that increased expression of EV-derived hemoglobin in neurons under hypoxia helps maintain mitochondrial homeostasis by sustaining mitochondrial membrane potential, thus aiding neurons in resisting hypoxic damage. This study clarified that the expression of exosomal hemoglobin in neurons contributes to the maintenance of mitochondrial homeostasis under stress and expanded the function of hemoglobin except transporting oxygen over a long range.

However, our research also had some limitations. First, we only clarified the hypoxic protective effect of hemoglobin on neuronal mitochondria in vitro and were unable to explain the inconsistency between the increase in neuronal hemoglobin expression and the decrease of mitochondrial activity observed in vivo. We speculated that some pathological changes may make hemoglobin, which is increased in response to hypoxia, unable to exert its mitochondrial protective effect, even more than making hemoglobin transform and exacerbating nerve damage. For example, previous studies reported large numbers of phosphorylated α-synuclein and hemoglobin aggregates in Lewy bodies in PD patients (*Killinger et al., 2022*), and we previously confirmed that chronic hypoxia increased

α-synuclein phosphorylation (*Li et al., 2022*). We therefore speculate that hemoglobin is increased in response to hypoxia and then phosphorylated by α-synuclein binding to stay in the cytoplasm, thus losing its mitochondrial protective ability and form pathological depositions, inducing cell death. However, further studies are needed to test this hypothesis.

In summary, we used the combined analysis of snRNA-seq and EVs-RNA-seq to examine the effects of hypoxia on interactions among different cell types in the brain and clarify the hypoxic response mechanism of non-neuronal EVs communication pathways to dynamically regulate hemoglobin levels in neurons. We revealed that increased levels of exosomal hemoglobin in neurons may help neurons to resist hypoxic injury by protecting mitochondrial function. These results clarify the molecular mechanism by which various non-neuronal cells interact via the EVs pathway to help neurons resist hypoxic injury and highlight the important role of hemoglobin in hypoxia-related nerve injury, thus providing a new perspective for the prevention and treatment of various hypoxic-ischemic brain diseases.

# Materials and methods

## Construction of chronic hypoxia model

A mouse model of chronic hypoxia was established in closed controlled hypoxic chambers (Ningbo Hua Yi Ning Chuang, China). Male C57BL/6J mice aged 2–3 months underwent a 28-day hypoxic cycle. Mice were maintained in a hypoxic chamber and the oxygen concentration was gradually decreased by 1% per day, from 21% $O_2$ to 7% $O_2$, and then maintained at 7% $O_2$ concentration for 14 days to establish a chronic hypoxic mouse model (H28d). Mice in the control group were maintained under 21% $O_2$ conditions (Con). All experimental animals were raised in a specific pathogen-free barrier environment, with normal temperature, humidity, and circadian rhythm, and given sufficient water and food. The study was approved by the Ethics Committee of Beijing Institute of Brain Disorders, Capital Medical University. All animal experiments were conducted in accordance with the National Institutes of Health Animal Research: Reporting of In Vivo Experiments (ARRIVE) guidelines (*Percie du Sert et al., 2020*).

## Behavioral tests

### Y-maze test

Y-maze tests were performed as described previously (*Maki et al., 2018*). The Y-maze consisted of three arms (40 × 4 × 9.5 cm, marked as A, B, and C, respectively) at angles of 120° to the central point. Each mouse was placed in the center of the starting arm and allowed to move freely in the maze for 8 min. The test was videotaped using a Nikon camera, manually recording the order of arm entry and calculating the percentage of alternations (calculated as [actual number of alternations/maximum number of alternations]×100%).

### Novel object preference test

Novel object preference tests were performed as described previously (*Lueptow, 2017*). The mice were pre-accustomed in an empty open-field box for three consecutive days (5 min each time). During training, the mice were exposed to two identical objects for 10 min. During the test, the mice were put back in the cage for 60 min before the test, a familiar object and a new object were added, and the mice were allowed to explore these freely for 5 min. The number of times the mice explored new and old objects was recorded using a Nikon camera. The discrimination index was calculated as ([number of new object explorations/total number of object explorations]×100). The cages and objects were cleaned with 75% ethanol to minimize any olfactory cues during the test.

### Open-field test

Open-field tests were performed as described previously (*Chow et al., 2019*). The mice were pre-accustomed in an open-field box (40 × 40 × 40 cm) for three consecutive days (5 min each time). During the test, the mice were placed in the open field and allowed to explore freely for 5 min. The movement time, movement distance, number of times through the central area, and residence time in the central area were recorded using a Nikon camera. The cages and objects were cleaned with 75% ethanol to minimize any olfactory cues during the test.

In the behavioral tests, mice with cognitive and motor disorders at baseline were excluded. All behavioral videos were analyzed blindly by researchers other than the operator.

## Neuropathological staining

### Nissl staining
Mouse brain tissue sections were fixed in 70% ethanol and dehydrated in graded ethanol solutions. Brain sections were permeabilized with xylene and incubated in 1% tar violet (Solarbio, China) for 30 min, washed with distilled water, and fixed in 70% alcohol. The slices were then dehydrated again in graded ethanol solutions and sealed with neutral glue.

### Fluoro-Jade B (FJB) staining
Mouse brain tissue sections were immersed in 100% ethanol for 3 min, followed by 70% ethanol for 1 min, ddH$_2$O for 1 min, and 0.006% potassium permanganate for 15 min. After rinsing in ddH$_2$O for 1 min, the sections were incubated in 0.001% FJB (Cell Signaling Technology, Beverly, MA, USA) staining solution for 30 min, immersed in xylene, dried, and sealed with an anti-quenching sealer containing DAPI.

### Golgi staining
Golgi–Cox staining was performed using a Golgi–Cox OptimStain Kit (Servicebio, China) to detect potential changes in the density and characteristics of neuronal dendritic spines. According to the instructions, whole brains were immersed in the dye solution for 14 days at room temperature in the dark. The tissue was then cut into 100 μm sections and photographed under a phase-contrast microscope. The types of dendritic spines were detected, and cells with clear dendritic spine resolution (>10 μm long) and straight end branches were selected for dendritic spine counting and analysis.

## Single-nuclear transcriptomics analysis of brain tissue

### Mouse brain tissue separation
Mice were euthanized by rapid neck dislocation and perfused with intracardiac pre-cooled phosphate-buffered saline (PBS). The brains were removed, dissected into 50–200 mg pieces, and quickly frozen in liquid nitrogen until the nucleus was extracted.

### snRNA-seq
Single-cell nuclear separation was performed as described previously (*Hodge et al., 2019*). Briefly, the frozen tissue was homogenized twice with a homogenizer (TIANZ, China), filtered through a 40 μm cell filter, centrifuged at 500 × *g* at 4°C for 5 min, and the nucleus was resuspended in cell suspension buffer. Cells/nuclei were counted using a blood cell counter, and the concentration was adjusted to 1500 nuclei per μL for library preparation.

The single-cell nuclear suspension was subjected to droplet generation, emulsion fragmentation, bead collection, reverse transcription, and cDNA amplification to generate a barcode library. The index library was constructed according to the manufacturer's instructions. The sequencing library was quantified using a Qubit ssDNA Assay Kit (Thermo Fisher Scientific, Waltham, MA, USA) and sequenced using BGI-Beijing's DIPSEQ-T7 sequencer.

### Analysis of snRNA-seq
CellRanger software was used to quantitatively analyze the gene expression of snRNA-seq data. The results were filtered using the R package Seurat (4.0.3), with removal of cells expressing 7000 genes and cells with mitochondrial gene content >25%. Unsupervised clustering was carried out using the Seurat package and the sample data were normalized using the NormalizeData function. The first 2000 highly variable genes were screened using the FindVariableGenes function. Uniform Manifold Approximation and Projection (UMAP) was used for dimensionality reduction and subsequent clustering. According to the clustering results, the corresponding gene markers were used for cell-type annotation. CellChat (v1.4.0) was used to infer and quantify cell–cell communication in brain cell clusters in the Con and H28d groups. The ligand–receptor interaction mouse database in CellChatDB (containing CellChat) was used as a reference for cell–cell interaction analysis.

## Transcriptome analysis of brain tissue EVs

### Isolation of brain tissue EVs

Brain tissue suspension after enzymatic hydrolysis was subjected to gradient centrifugation at 300, 2000, and 10,000 × $g$ at 4°C, and filtered through a 0.22 μm sieve to remove all cell debris and apoptotic bodies. The collected supernatant was ultracentrifuged at 4°C, 110,000 × $g$ for 2 h to obtain a precipitate containing EVs. The precipitate was resuspended in PBS and further purified using an Exosupur column (Echobiotech, China). The fraction was concentrated to a final volume of 200 μL using a 100 kDa cut-off Amicon Ultra rotary filter (Merck, Germany). Details of the process are provided in the EV-TRACK protocol (EV220325) (*Van Deun et al., 2017*).

### Western blots of EVs

The collected EV supernatant was denatured in sodium dodecyl sulfate buffer and detected by western blot. The purified products were characterized as EVs by incubation with primary antibodies for the EV-positive markers CD9 (Abcam, ab92726), HSP70 (Abcam, ab181606), and TSG101 (Abcam, ab125011), and the EV-negative marker calnexin (Proteintech, 10427-2-AP).

### Transmission electron microscopy of EVs

The collected EV supernatant was fixed with 2% paraformaldehyde solution for 20 min. Sample droplets were then incubated on a copper mesh containing a biological support membrane for 5–10 min, washed twice with ddH$_2$O, stained with 2% uranyl acetate for 5 min, and washed twice with ddH$_2$O. The samples were then dried thoroughly using an infrared baking lamp, and then observed and photographed with a transmission electron microscope (Hitachi Ltd., Tokyo, Japan).

### Nanoparticle tracking analysis (NTA) of EVs

The purified vesicle suspension was observed using a ZetaView PMX 110 equipped with a 405 nm laser (Particle Metrix, Meerbusch, Germany) to determine the number and size of the separated particles. A 60 s video was recorded at a speed of 30 frames/s, and the particle motion was analyzed using NTA software (ZetaView 8.02.28).

### Analysis of EVs-RNA-seq

A total of 250 pg to 10 ng RNA per sample was used as input material for sequencing libraries (1–500 ng for small RNA libraries) using the SMARTer Stranded Total RNA-Seq Kit V2 (Takara Bio USA, Inc) following the manufacturer's recommendations, and index codes were added to attribute sequences to each sample.

RNA per sample was used as input material for the RNA sample preparations. Sequencing libraries were generated using a QIAseq miRNA Library Kit (QIAGEN, Frederick, MD, USA) following the manufacturer's recommendation, and index codes were added to attribute sequences to each sample. Reverse transcription (RT) primers with unique molecular indices were introduced to analyze the quantification of mRNA expression during cDNA synthesis and PCR amplification. The library quality was finally assessed using an Agilent Bioanalyzer 2100 and quantitative PCR (qPCR). Clustering of the index-coded samples was performed using an acBot Cluster Generation System using TruSeq PE Cluster Kitv3-cBot-HS (Illumina, San Diego, CA, USA) according to the manufacturer's instructions. After cluster generation, the library preparations were sequenced on an Illumina Hiseq platform and paired-end reads were generated.

Raw data (raw reads, fastq format) were first processed using in-house perl scripts. In this step, clean reads were obtained by removing reads containing adapters, poly-N reads, and low-quality reads. At the same time, the Q20, Q30, GC-content, and sequence duplication level of the clean data were calculated. All downstream analyses were based on high-quality clean data. Paired-end clean reads were aligned to the reference genome GRCh38 using TopHat2/Bowtie2. Mapped reads were used to quantify the gene expression level and for differential expression analysis.

Stringtie was used to calculate fragments per kilobase of exon per million fragments mapped (FPKMs, calculated based on the length of the fragments and reads count mapped to this fragment) of coding genes in each sample. FPKMs were computed by summing the FPKMs of transcripts in each gene group. Differential expression was determined between the Con and H28d groups.

## Combined analysis of brain tissue EV transcriptome and single-cell transcriptome

First, the online tool CIBERSORTx for deconvolution was used to create a signature matrix from snRNA-seq data (**Newman et al., 2019**). The parameter 'Min. Expression' was set to zero and 'Replicates' was set to 100. In summary, this combined analysis method was used to clarify the origin of EVs containing different mRNAs.

## Real-time PCR

Total RNA was extracted from mouse cerebral cortex using a SteadyPure universal RNA extraction kit (Accurate Bio, China). The RNA was then reverse transcribed into cDNA using an Evo M-MLV reverse transcription premix kit (Accurate Bio). Real-time PCR was carried out using an ABI real-time PCR system (Life Technologies, USA) and SYBR Green Pro TaqHS premix qPCR kit III (Accurate Bio), according to the manufacturer's instructions. The following primers were used:

> Mouse *Actb*, Forward: 5'-GGCTGTATTCCCCTCCATCG-3',
> Reverse: 5'-CCAGTTGGTAACAATGCCATGT-3';
> Mouse *Hba-a1*, Forward: 5'-GCCCTGGAAAGGATGTTTGC-3',
> Reverse: 5'-TTCTTGCCGTGACCCTTGAC-3';
> Human *RPL19*, Forward: 5'-CTAGTGTCCTCCGCTGTGG-3',
> Reverse 5'-AAGGTGTTTTTCCGGCATC-3';
> Human *HBA1*, Forward: 5'-TTCTGGTCCCCACAGACTCA-3',
> Reverse: 5'-CTAGTGTCCTCCGCTGTGG-3'.

## Western blot

Proteins were extracted from cells or tissues using RIPA buffer (Solarbio, China), and total protein concentrations were quantified using a Pierce BCA Protein Assay Kit (Thermo Fisher Scientific). Samples for gel electrophoresis were prepared by boiling in loading buffer for 10 min. The protein lysates were then analyzed by sodium dodecyl sulfate-polyacrylamide gel electrophoresis, and immunoblotted onto a polyvinylidene fluoride membrane. The membrane was blocked with 5% skim milk at room temperature for 1 h, washed with TBST, and then incubated with the specified primary antibody at 4°C overnight. The primary antibodies included β-actin (HuaBio, EM21002, 1:10,000), Hba (Abcam, ab92492, 1:1000), NDUFB8 (Proteintech, 14794-1-AP, 1:5000), and Citrate synthase (Proteintech, 16131-1-AP, 1:2000).After incubation, the membrane was washed three times and then incubated with the secondary antibody at room temperature for 1 h, including IRDye 680RD goat anti-mouse IgG (H+L) (LI-COR, 926-68070), IRDye 800CW goat anti-rabbit IgG (H+L) (LI-COR, 926-32211). The membrane was scanned using a detection system (Odyssey, USA) and the band intensity was normalized to β-actin. Statistical analysis was performed using ImageJ and GraphPad software.

## Immunofluorescence

Mouse brain sections were placed in citric acid buffer (pH 6.0), boiled for 10 min for antigen retrieval, and then washed three times with PBS for 5 min each time. The brain slices were then placed in 0.2% PBS-Triton and incubated at room temperature for 30 min, followed by 5% bovine serum albumin–PBS solution, and then blocked at room temperature for 2 h. The slices were then incubated overnight in a primary antibody diluent at 4°C in the refrigerator. The following primary antibodies were used: Hba (Abcam, ab92492; Proteintech, 14537-1-AP) and MAP2 (Abcam, ab32454). Brain slices were washed three times with PBS for 5 min each time, and then incubated with fluorescent secondary antibodies in the dark at room temperature for 1 h. The following secondary antibodies were used: goat anti-mouse IgG (H+L) highly cross-adsorbed secondary antibody Alexa Fluor 488 (Invitrogen/ThermoFisher), and goat anti-rabbit IgG (H+L) highly cross-adsorbed secondary antibody Alexa Fluor 594 (Invitrogen/ThermoFisher). Nuclei were stained with DAPI before mounting (Sigma, D9542). Images were observed and captured by confocal microscopy.

## Cell culture

### Primary neurons

Primary cortical neuron cultures were prepared from Sprague–Dawley rat embryos at embryonic days 16–18. The cortex was dissected and separated mechanically by blowing through a straw to create a single-cell suspension, washed, and counted. The single-cell suspension was centrifuged at $1000{\times}g$ for 5 min and resuspended in DMEM by blowing, as above. The cell suspension was passed through a 40 μm filter. The cells were diluted to the required density and then transferred to a polylysine-coated plate and cultured in a 5% $CO_2$ incubator at 37°C for 4 h. The DMEM was then replaced with Neurobasal medium containing B27, glutamine, and penicillin–streptomycin. Half of the medium was replaced every 3 days after cell adhesion.

### Cell lines

The human neuroblastoma cell line (SH-SY5Y) and human brain microvascular endothelial cell line (hcMEC/D3) were purchased from the American Center for Type Culture Collection (ATCC, USA). SH-SY5Y and hcMEC/D3 cells were cultured at 37°C in DMEM supplemented with 10% fetal bovine serum, penicillin (100 U/mL), and streptomycin (100 μg/mL) in a humidified 5% $CO_2$/95% air incubator. The cells were subjected to hypoxia in a three-gas incubator (Ningbo Hua Yi Ning Chuang, China) rinsed with a gaseous mixture of 1% $O_2$, 5% $CO_2$, and balanced $N_2$. The cells were inoculated and cultured under normoxia for 12 h to adhere, and then transferred to a hypoxic incubator for a specified time. All cell culture reagents were from Thermo Fisher Scientific.

### Isolation of EVs from cell culture medium

EVs isolation from cell culture medium were performed as described previously (*Zhang et al., 2023*). After treatment of hcMEC/D3 cells, the culture medium was collected and centrifuged at $300{\times}g$ for 5 min at room temperature to remove cell debris, followed by $3000{\times}g$ at 4°C for 10 min to remove apoptotic bodies, and then filtered with a 0.22 μm filter membrane. The supernatant was then centrifuged at $10,000{\times}g$ for 30 min to remove large microbubbles (>500 nm). The supernatant was ultracentrifuged at $110,000{\times}g$ at 4°C for 70 min, and the EVs in the precipitate were collected and re-suspended in PBS.

### EV uptake detection

Dil solution (5 μM; Invitrogen/Thermo Fisher) was added to PBS containing EVs and incubated for 30 min. Excess dye in the labeled EVs was removed by ultracentrifugation at $110,000{\times}g$ at 4°C for 1 h. The EVs were then washed three times by re-suspending the precipitate in PBS, and the final washing was re-suspended in PBS. PBS solution containing Dil-EVs was added to a culture dish containing SH-SY5Y cells, and Dil-EV uptake by the cells was observed under a confocal microscope after 24 h of culture.

### Cell viability

Cells were inoculated into 96-well plates at a density of 2000–3000 cells per well. CCK-8 cell viability assay (New Cell and Molecular Biotech, China) was carried out as follows: DMEM (1:10) solution was added to each well of the cell plate containing cells and samples and incubated at 37°C for 3 h. The absorbance (optical density) of each well was then measured at 450 nm using a microplate reader (SpectraMax i3, Molecular Devices, USA).

### Cytotoxicity detection

Cytotoxicity was measured by lactate dehydrogenase (LDH) assay (Roche, USA). The powder was dissolved in $ddH_2O$ and fully mixed to prepare a catalytic solution, followed by adding 250 μL of this catalytic solution to the dyeing solution (11.25 mL) and mixing to prepare the LDH reaction solution. Cell supernatant (100 μL) from each group was then added to a new 96-well plate and 100 μL LDH reaction solution was added and incubated at room temperature in the dark for 30 min. After incubation, 50 μL of termination solution was added to each well and gently mixed for 10 min, and the absorbance of each well at 490 nm was measured using a microplate reader.

## Mitochondrial viability detection

Mitochondria in the cultured cells were labeled and their activity was detected using MitoTracker dye (Invitrogen/Thermo Fisher), and the mitochondrial membrane potential was detected using JC-1 dye (Sigma-Aldrich, USA). MitoTracker dye was diluted to 50 nM with DMEM before staining, and JC-1 dye was diluted to 10 µg/mL with DMEM before staining, and stored in the dark at 4°C until use. After removing the original medium at the designated treatment end point, the cultured cells were digested and centrifuged at 1000×$g$ for 5 min in 1.5 mL centrifuge tubes and then resuspended in DMEM with dye and incubated for 20 min. The dye-containing medium was then removed by centrifugation at 1000×$g$ for 5 min, and the cells were washed twice with PBS, and then resuspended in PBS. The fluorescence intensity of each sample was detected using a BD LSRFortessa SORP flow cytometer (Becton, Dickinson and Company, USA). Follow-up analysis was performed using FlowJo.

## Cell death detection

Propidium iodide (PI)/Hoechst staining was used to detect primary neuronal cell death. The culture medium was discarded and the cells were washed three times with PBS, followed by incubation with PI (4 mM) and Hoechst (0.5 mg/mL) (Sigma-Aldrich) at 37°C for 10 min, and washed with PBS for 5 min. Images were observed and captured by confocal microscopy.

## Exchange of endothelial cell-conditioned medium

Medium exchange was performed to determine if EC-conditioned medium could exert neuroprotective effects. hcMEC/D3 cells were subjected to hypoxia and the culture medium was then collected. Cell debris and apoptotic bodies were removed by gradient centrifugation and filtered with a 0.22 µm filter membrane to obtain EC-conditioned medium. The conditioned medium was added to SH-SY5Y cells that had been adherent for 12 h, and subsequent hypoxia treatment and cell detection were performed.

EV removal and medium-exchange experiments were used to determine if the EVs in EC-conditioned medium could exert neuroprotective effects. The collected conditioned medium was subjected to ultracentrifugation to remove EVs, followed by medium exchange and subsequent experiments.

*HBA1* RNAi plasmid transfection combined with medium-exchange experiments was used to determine if *HBA1* mRNA in EVs from EC-conditioned medium had a neuroprotective role. The hcMEC/D3 cells were transfected with *HBA1* RNAi plasmid mixed with Lipofectamine 3000 and P3000 (Invitrogen/Thermo Fisher) in advance for 24 h, followed by hypoxia and conditioned medium collection, and medium exchange and subsequent experiments were performed.

## Statistical analysis

Data storage, recording, and statistical analysis were carried out using Excel and GraphPad Prism 10.0 software. The image data were analyzed using ImageJ. The Shapiro–Wilk tests and the Kolmogorov–Smirnov tests were used to check the normality of the variables. Unpaired $t$-tests (two-tailed) were used to compare differences between two sets of normally distribution data with equal variance, otherwise, non-parametric tests were used. Differences among multiple groups were analyzed by one-way/two-way ANOVA followed by Tukey's *post hoc* test. Image data were analyzed by ImageJ. Data were expressed as mean ± standard error (SEM), and $p < 0.05$ was considered statistically significant.

# Acknowledgements

This research was supported by the National Natural Science Foundation of China (grant number: 32100925), the Beijing Nova Program (grant number: Z211100002121038, 20230484436), the Chinese Institutes for Medical Research, Beijing (grant number: CX23YQ01), and Beijing-Tianjin-Heibei Basic Research Cooperation Project (grant number: 22JCZXJ00190).

## Additional information

### Funding

| Funder | Grant reference number | Author |
| --- | --- | --- |
| National Natural Science Foundation of China | 32100925 | Jia Liu |
| Beijing Nova Program | Z211100002121038 | Jia Liu |
| Beijing Nova Program | 20230484436 | Jia Liu |
| Chinese Institutes for Medical Research | CX23YQ01 | Jia Liu |
| Beijing-Tianjin-Hebei Basic Research Cooperation | 22JCZXJ00190 | Jia Liu |

The funders had no role in study design, data collection and interpretation, or the decision to submit the work for publication.

### Author contributions
Zhengming Tian, Conceptualization, Data curation, Formal analysis, Investigation, Visualization, Methodology, Writing – original draft, Writing – review and editing; Yuning Li, Data curation, Investigation, Methodology; Feiyang Jin, Investigation, Methodology; Zirui Xu, Yakun Gu, Mengyuan Guo, Qianqian Shao, Yingxia Liu, Hanjiang Luo, Yue Wang, Suyu Zhang, Chenlu Yang, Xin Liu, Methodology; Xunming Ji, Conceptualization, Formal analysis, Supervision, Funding acquisition, Validation, Project administration, Writing – review and editing; Jia Liu, Conceptualization, Data curation, Formal analysis, Supervision, Funding acquisition, Validation, Investigation, Methodology, Writing – original draft, Project administration, Writing – review and editing

### Author ORCIDs
Zhengming Tian ![ORCID] https://orcid.org/0000-0001-6183-4701
Xunming Ji ![ORCID] https://orcid.org/0000-0002-0527-2852
Jia Liu ![ORCID] https://orcid.org/0000-0001-6711-3841

### Ethics
C57BL/6J wild-type male mice used in the study were purchased from SPF (Beijing) Biotechnology Co., Ltd. Mice were maintained on a 12 light/12 dark cycle. All animal procedures were approved by the Animal Care and Use Committee of Capital Medical University (permit no. AEEI-2022-073) and performed according to the National Institutes of Health Guide for the Care and Use of Laboratory Animals. All efforts were made to minimize animal suffering and the number of animals used. All animal experiments complied with the ARRIVE guidelines.

Joint public review https://doi.org/10.7554/eLife.99986.3.sa1
Author response https://doi.org/10.7554/eLife.99986.3.sa2

## Additional files

### Supplementary files
MDAR checklist

### Data availability
EVs-RNA-seq data have been deposited in the Genome Sequence Archive (GSA) under accession number CRA025842 and are publicly accessible at https://ngdc.cncb.ac.cn/gsa/browse/CRA025842. Single-nucleus RNA-seq data have been deposited in the Dryad Digital Repository and are available at https://doi.org/10.5061/dryad.r7sqv9sq4. All other data generated or analyzed during this study are included in the manuscript and supporting files. Source data files have been provided for all main figures.

The following datasets were generated:

| Author(s) | Year | Dataset title | Dataset URL | Database and Identifier |
|---|---|---|---|---|
| Tian Z, Li Y, Jin F, Xu Z, Gu Y, Guo M, Shao Q, Liu Y, Luo H, Wang Y, Zhang S, Yang C, Liu X, Ji X, Liu J | 2025 | Brain-derived exosomal hemoglobin transfer contributes to neuronal mitochondrial homeostasis under hypoxia | https://doi.org/10.5061/dryad.r7sqv9sq4 | Dryad Digital Repository, 10.5061/dryad.r7sqv9sq4 |
| Tian Z, Li Y, Jin F, Xu Z, Gu Y, Guo M, Shao Q, Liu Y, Luo H, Wang Y, Zhang S, Yang C, Liu X, Ji X, Liu J | 2025 | Exosomal RNA-seq of mouse brain under chronic hypoxia | https://ngdc.cncb.ac.cn/gsa/browse/CRA025842 | Genome Sequence Archive (GSA), CRA025842 |

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
