## [Editor Report · eLife Assessment]

This **valuable** article analyzes the role of endogenous CNS hemoglobin in protecting mitochondrial homeostasis in hypoxic conditions. The work is **solid** and opens the doors to future work in this field. However, it leaves many questions open regarding CNS-specific ischemia/hypoxia that should be considered in future work. In particular, a whole-body hypoxia model may liberate exosomes from other hypoxic organs, which may contribute to the protective effect. Overall, this work has the potential to be of broad interest to the neuroscience and hypoxia communities.

---

## [Referee Report · Joint public review]

Summary:

This study investigates the hypoxia rescue mechanisms of neurons by non-neuronal cells in the brain from the perspective of exosomal communication between brain cells. Through multi-omics combined analysis, the authors revealed this phenomenon and logically validated this intercellular rescue mechanism under hypoxic conditions through experiments. The study proposed a novel finding that hemoglobin maintains mitochondrial function, expanding the conventional understanding of hemoglobin. This research is highly innovative, providing new insights for the treatment of hypoxic encephalopathy.

Overall, the manuscript is well organized and written, however, the authors have only partially answered the reviewers comments.

---

## [Author Response]

The following is the authors’ response to the original reviews

**Reviewer #1 (Public Review):**
Summary:This study investigates the hypoxia rescue mechanisms of neurons by non-neuronal cells in the brain from the perspective of exosomal communication between brain cells. Through multi-omics combined analysis, the authors revealed this phenomenon and logically validated this intercellular rescue mechanism under hypoxic conditions through experiments. The study proposed a novel finding that hemoglobin maintains mitochondrial function, expanding the conventional understanding of hemoglobin. This research is highly innovative, providing new insights for the treatment of hypoxic encephalopathy.Overall, the manuscript is well organized and written, however, there are some minor/major points that need to be revised before this manuscript is accepted.

We thank the reviewer for the detailed analysis of our study. Please find our answers to the points raised by the reviewer below.

Major points:(1) Hypoxia can induce endothelial cells to release exosomes carrying hemoglobin, however, how neurons are able to actively take up these exosomes? It is possible for other cells to take up these exosomes also? This point needs to be clarified in this study.

We sincerely appreciate the reviewer’s valuable comments. Regarding the question of how neurons actively uptake extracellular vesicles (EVs) carrying hemoglobin mRNA, existing studies suggest that EVs can enter cells via three main pathways: direct fusion, receptor-mediated endocytosis, and phagocytosis (PMID: 25288114). Our experimental results show that neurons are able to actively uptake EVs from endothelial cells without any treatment, and hypoxic conditions did not significantly increase the uptake of endothelial EVs by neurons (Fig. 5A and I). As for the specific uptake mechanism, there is currently no definitive conclusion. Some studies have found that hypoxic-ischemic injury may induce neurons to upregulate Cav-1, which could enhance the uptake of endothelial-derived EVs via Cav-1-mediated endocytosis (PMID: 31740664), but this mechanism still requires further validation.

Regarding whether other cell types also take up these EVs, we focused on neurons based on existing literature and our own data, which show that the increased hemoglobin in the brain under hypoxic conditions is primarily found in neurons (Fig. 4H-J, PMID: 19116637). Moreover, we observed that, under hypoxic conditions, almost all non-neuronal supporting cells in the brain transcribe hemoglobin in large amounts and release it via EVs (Fig. 3J). Furthermore, we would like to emphasize that although neurons do not transcribe hemoglobin, we observed substantial expression of hemoglobin within neurons. This suggests that it may serve as an important protective mechanism for the brain. Therefore, the focus of our study is on the protective effect of EVs carrying hemoglobin mRNA on neurons, and the uptake by other cell types was not explored. We greatly appreciate the reviewer’s question, and we believe this is an intriguing avenue for further investigation. This could provide new insights for interventions in hypoxic brain injury, and we plan to delve into this topic in future studies.

(2) The expression of hemoglobin in neurons is important for mitochondrial homeostasis, but its relationship with mitochondrial homeostasis needs to be further elucidated in the study.

We sincerely appreciate the reviewer’s valuable comments. We fully agree with the importance of hemoglobin expression in neurons for mitochondrial homeostasis. In this study, we have confirmed through in vitro experiments that when neurons are treated with conditioned medium from endothelial cells, they exhibit increased hemoglobin expression. This, in turn, enhances their resistance to hypoxia by restoring mitochondrial membrane potential and increasing mitochondrial numbers, thereby effectively improving neuronal viability. Notably, this protective effect disappears when EVs are removed from the endothelial-conditioned medium or when hemoglobin in endothelial cells is disrupted, further supporting the notion that endothelial cells transfer hemoglobin via EVs, helping neurons express hemoglobin under hypoxic conditions and exert protective effects.

In summary, hemoglobin primarily helps maintain mitochondrial membrane potential, thereby supporting the restoration of energy metabolism and production under hypoxic conditions, which effectively improves the neuronal resistance to hypoxia. Although we were unable to explore the specific mechanisms of hemoglobin’s role in mitochondrial homeostasis in detail within this study, we recognize the importance of this aspect and plan to further investigate how hemoglobin regulates mitochondrial homeostasis and function in neurons in future research.

Once again, we greatly appreciate the reviewer’s insightful comments. We will continue to optimize our research direction and look forward to further elucidating these important biological mechanisms in future studies.

Minor points:(1) In Figures 1-3, the authors use "Endo" to represent endothelial cells, while in Figures 4-7, the abbreviation "EC" is used. Please standardize the format.

Thank you for the reviewer’s suggestion. We will use “EC” consistently to refer to endothelial cells throughout the manuscript to ensure uniformity.

(2) In all qPCR statistical results, please italicize the gene names on the axis.

Thank you for the reviewer’s valuable suggestion. We will make sure to italicize the gene names on the axis in all qPCR statistical results to adhere to the formatting requirements.

(3) In the Western blot result of Figure 3C, what type of cell-derived exosomes does the Control group represent, and why can it be used as a control group for brain-derived exosomes?

Thank you for the reviewer’s insightful question. In Fig. 3C, the control group (Control) represents the cell lysate sample, which serves as a positive control in the EVs Western blot analysis. In this experiment, the positive control is primarily used to validate the specificity of the antibody and the accuracy of the experimental procedure. We used cell lysate as the control to confirm that the antibody can detect EV-associated markers in the cell lysates, thus providing a comparative basis for the identification of brain-derived EVs.

(4) In Figure 4F, the morphology of hemoglobin in the Con group and the H28d group is not entirely consistent with Figure 4H. Is this difference due to different experimental batches?

Thank you for the reviewer’s careful observation. The observed difference may indeed be due to variations between different experimental batches. To ensure consistency of the results, we have updated the representative immunofluorescence images, which are now presented in Fig. 4H.

(5) Supplement the transcription and expression levels of hemoglobin in neurons under different treatment conditions after medium exchange with exosome removal and medium exchange after HBA1 interference.

Thank you for the reviewer’s valuable suggestions. We have added the experimental data regarding the exchange of culture medium after the removal of EVs. As shown in Fig. S6, the endothelial-derived medium without EVs does not enhance the hemoglobin levels in neurons under hypoxic conditions. Additionally, we have included the detection results of hemoglobin expression in neurons after *HBA1* interference, as shown in Fig. S7E-F. The results indicate that the culture medium derived from HBA1-interfered endothelial cells also fails to help neurons increase hemoglobin expression under hypoxic conditions.

(6) Figure S3 should be split to separately explain the increased exosome release induced by hypoxia, the non-toxic effect of endothelial cell culture medium on neurons, and the successful screening of the HBA1 interference plasmid.

Thank you for the reviewer’s suggestions. Based on your feedback, we have split the original Fig. S3 into multiple parts to more clearly present the different experimental results. Specifically, the results of hypoxia-induced EVs release increase have been updated in Fig. S4, the non-toxic effects of endothelial cell culture medium on neurons are shown in Fig. S5, and the successful screening of the *HBA1* interference plasmid is presented in Fig. S7.

(7) Regarding the extracellular vesicles/exosomes, it should be expressed consistently in the whole manuscript.

Thank you for the reviewer’s reminder. We will ensure that the term “extracellular vesicles” is used consistently throughout the manuscript.

(8) In lines 70 and 80, the O2 should be changed to "O_2_".

Thank you for the reviewer’s careful observation. We have corrected the formatting of “O2” to “O₂” in lines 70 and 80.

We would like to thank the Reviewer for taking the time to thoroughly examine our work, for their helpful feedback that has significantly contributed to improving our manuscript, and for their kind and encouraging words.

**Reviewer #2 (Public Review):**
Summary:This is an interesting study with a lot of data. Some of these ideas are intriguing. But a few major points require further consideration.We thank the reviewer for the detailed assessment of our study and pinpointing its current weaknesses. Please find our answers to all comments below.Major points:(1) What disease is this model of whole animal hypoxia supposed to mimic? If one is focused on the brain, can one just use a model of focal or global cerebral ischemia?

Thank you for the reviewer’s insightful question. The chronic hypoxia model we employed is designed to mimic the multi-organ damage caused by systemic hypoxia, which is relevant to clinical conditions such as high-altitude hypoxia, chronic obstructive pulmonary disease, and acute hypoxic brain injury. In contrast to focal or global cerebral ischemia models, the focus of our study is on how the brain, under extreme systemic hypoxia, utilizes endothelial cell-derived extracellular vesicles (EVs) to transfer hemoglobin mRNA, thereby protecting neurons and aiding the brain’s response to hypoxia-induced damage.

We understand the reviewer’s concern that focal or global ischemia models are typically used to simulate localized brain hypoxia or ischemic injury. However, the core of our research is to explore the brain’s overall adaptive mechanisms under systemic hypoxic conditions. By using a systemic hypoxia model, we can more comprehensively simulate the effects of global hypoxia on the brain and uncover how the brain engages specific molecular mechanisms for self-protection. This approach offers a novel perspective on brain hypoxic-ischemic diseases and holds potential clinical applications, particularly in the study of stroke, vascular cognitive impairment and dementia (VCID), and related conditions.

Additionally, we have observed that hemoglobin significantly increases in the brain in an animal model of focal ischemia (as shown in Author response image 1 below). This finding further supports the idea that hemoglobin upregulation may be a universal protective mechanism for the brain’s response to hypoxic damage. While this part of the research is still ongoing, preliminary results suggest that both systemic hypoxia and focal ischemia might trigger protective effects through hemoglobin regulation.

**Author response image 1. sa2fig1:** The expression level of Hba-a1 in the brain of VCID mouse.

Therefore, the core of our study is to elucidate the brain’s self-protection mechanisms under systemic hypoxia, rather than focusing solely on cerebral ischemia models. We believe this approach provides new insights into the prevention and treatment of brain hypoxic-ischemic diseases, with significant clinical application potential.

In light of this, we have added a related discussion to the manuscript, clearly explaining the rationale for choosing the systemic hypoxia model. The updated content can be found on P11, Line 13-21 as follows: “To investigate this phenomenon, we employed a chronic hypoxia model in which mice were exposed to 7% oxygen for 28 days. This model aims to mimic systemic hypoxia-induced multi-organ damage, a condition observed in diseases such as high-altitude hypoxia, chronic obstructive pulmonary disease, and acute hypoxic brain injury. The primary goal of this model is to explore how the brain adapts under extreme low-oxygen conditions and employs specific mechanisms to protect itself from hypoxia-induced damage. This approach provides valuable insight into diseases related to hypoxic-ischemic injury in the brain, including stroke and vascular dementia, offering a novel perspective for potential clinical applications.”

(2) If this model subjects the entire animal to hypoxia, then other organs will also be hypoxic. Should one also detect endothelial upregulation and release of extracellular vesicles containing hemoglobin mRNA in non-CNS organs? Where do these vesicles go? Into blood?

Thank you for the reviewer’s valuable feedback. Indeed, in a whole-body hypoxia model, other organs are also affected by hypoxia. Therefore, future research may need to investigate the upregulation of endothelial cells in organs other than the central nervous system, as well as the release of EVs containing hemoglobin mRNA from these organs. However, in this study, we isolated EVs from the brain tissue in situ following perfusion with physiological saline, a method that effectively eliminates the influence of EVs from blood or other organs. As a result, our primary focus was on studying how EVs released by brain endothelial cells are actively taken up by neurons to exert neuroprotective effects. The potential for these EVs to enter the bloodstream and their subsequent fate is indeed a topic worthy of further investigation. Future research could offer new insights into the cross-organ effects of systemic hypoxia.

(3) What other mRNA are contained in the vesicles released from brain endothelial cells?

Thank you for the reviewer’s valuable suggestions. We have further analyzed EVs derived from brain endothelial cells, and in addition to hemoglobin mRNA, these EVs also contain a variety of other mRNAs, including *Vwf, Hbb-bt*, *Hba-a1*, *Hbb-bs*, *Hba-a2*, *Acer2*, *Angpt2*, *Ldha*, *Gm42418*, *Slc16a1*, *Cxcl12*, *B2m*, *Ctla2a*, *Ccnd1*, and *Hmgcs2* (Log2FC > 1.2). The biological processes associated with these mRNAs primarily involve: cell-substrate adhesion, regulation of cellular amide metabolic process, negative regulation of cell migration, negative regulation of cell motility, and negative regulation of cellular component movement. These processes may be closely related to the neuroprotective effects of endothelial cell EVs in a hypoxic environment, especially in terms of regulating cell behavior and maintaining cell structure and function. Additionally, these EVs contain multiple key factors associated with intracellular metabolism, movement, and migration, which may collectively influence neuronal function and survival. Notably, our study also found that mRNA of various hemoglobin subunits ranks among the top five in terms of abundance in the mRNA secreted by hypoxic endothelial EVs, further emphasizing the importance of hemoglobin mRNA in endothelial-derived EVs. Therefore, future research may explore the functions of these mRNAs and reveal how they act in concert to protect neurons from hypoxia-induced damage.

We have updated and added these results in Fig. S4, and have further elaborated on the findings in the revised figure. Once again, we thank the reviewer for the attention and valuable suggestions regarding our work.

(4) Where do the endothelial vesicles go? Only to neurons? Or to other cells as well?

Thank you for the reviewer’s important question. As previously mentioned, the focus of this study is to investigate how EVs carrying hemoglobin mRNA influence neuronal function. Through a combined analysis of single-cell transcriptomics and EV transcriptomics from brain tissue, we found that, besides neurons, almost all types of supportive cells in the brain and their secreted EVs contain a significant amount of hemoglobin mRNA (Fig. 3J, 4B). Notably, although neurons do not transcribe hemoglobin mRNA themselves, under hypoxic conditions, neurons significantly increase hemoglobin expression, resulting in a phenomenon where the transcription and expression levels of hemoglobin in neurons are inconsistent. This phenomenon has been observed both in our study and others (Fig. 4H-J, PMID: 19116637). This observation led us to focus on the active uptake of EVs by neurons and the potential neuroprotective effects they might bring.

Regarding whether other cell types uptake these EVs and their potential functions, although our current research is focused on neurons, this is indeed an important area for further investigation. Given that non-neuronal supportive cells may also transfer hemoglobin mRNA via EVs under hypoxic conditions, future research will further explore the uptake of EVs by different cell types and their roles in hypoxic adaptation.

We are particularly interested in the hemoglobin expression in neurons under hypoxic conditions and consider neurons to be the primary expressers of hemoglobin, providing a new perspective for exploring the neuroprotective role of hemoglobin. We plan to delve deeper into these issues in future studies.

(5) Neurons can express endogenous hemoglobin. Is it useful to subject neurons to hypoxia and then see how much the endogenous mRNA goes up? How large is the magnitude of endogenous hemoglobin gene upregulation compared to the hypothesized exogenous mRNA that is supposed to be donated from endothelial vesicles?

Thank you for the reviewer’s valuable question. We have observed that, in the absence of treatment with endothelial cell-derived conditioned medium, there is no significant change in the transcription and expression levels of endogenous hemoglobin in neurons under hypoxic conditions (Fig. 5I, 6C-D). However, when neurons were treated with endothelial cell-conditioned medium, under the same hypoxic conditions, the transcription levels of hemoglobin increased by approximately 1.2-fold, and the expression levels increased by approximately 3.8-fold (Fig. 6B-D). Additionally, we have added pre-treatment experiments involving EVs depletion from the endothelial culture medium and HBA interference. The results show that, after these two pre-treatments, the conditioned medium lost its ability to enhance the transcription and expression of hemoglobin in neurons under hypoxic conditions (Fig. S6, S7D-F), further emphasizing the important role of endothelial EVs in this process. This finding indicates that endothelial-derived EVs significantly promote hemoglobin expression in neurons, and this effect is far greater than the upregulation of endogenous hemoglobin in neurons. Therefore, while neurons can express endogenous hemoglobin, exogenous hemoglobin significantly enhances its expression, which may help neurons tolerate the hypoxic environment and provide additional protection.

(6) Finally, it may be useful to provide more information and data to explain how the expression of this exogenous endothelial-derived hemoglobin binds to neuronal mitochondria to alter function.

Thank you for the reviewer’s valuable suggestion. As we previously mentioned, hemoglobin plays a protective role in neurons by maintaining mitochondrial membrane potential, helping neurons restore energy metabolism and energy production under hypoxic conditions. We fully agree on the importance of this research direction. Several studies have shown that when hemoglobin is expressed in neurons, it predominantly localizes to mitochondria, which aligns with the physiological process of heme synthesis within mitochondria (PMID: 23187133). Furthermore, in the brains of Parkinson’s disease patients, the localization of hemoglobin in neuronal mitochondria is altered compared to normal conditions (PMID: 27181046). Therefore, the interaction between hemoglobin and mitochondria plays a crucial role in neuronal function.

Although existing research indicates the role of hemoglobin in neuronal mitochondria, studies in this area remain limited. We plan to further investigate how hemoglobin binds to mitochondria and its specific effects on mitochondrial function in our future work. We believe that a deeper understanding of this mechanism will provide essential theoretical insights into the effects of hypoxia on neurons and offer new potential strategies for neuroprotective therapies.

We would like to thank the Reviewer for taking the time to thoroughly examine our work, for their helpful feedback that has significantly contributed to improving our manuscript, and for their kind and encouraging words.